# Seven naturally variant loci serve as genetic modifiers of *Lamc2^jeb^* induced non-Herlitz junctional Epidermolysis Bullosa in mice

**Thomas J. Sproule**[1]*, **Vivek M. Philip**[1], **Nabig A. Chaudhry**[1], **Derry C. Roopenian**[1], **John P. Sundberg**[1,2]

**1** The Jackson Laboratory, Bar Harbor, Maine, United States of America, **2** Department of Dermatology, Vanderbilt University Medical Center, Nashville, Tennessee, United States of America

* tom.sproule@jax.org

**Data Availability Statement:** All relevant data are within the paper and its Supporting information files.

## Abstract

Epidermolysis Bullosa (EB) is a group of rare genetic disorders that compromise the structural integrity of the skin such that blisters and subsequent erosions occur after minor trauma. While primary genetic risk of all subforms of EB adhere to Mendelian patterns of inheritance, their clinical presentations and severities can vary greatly, implying genetic modifiers. The *Lamc2^jeb^* mouse model of non-Herlitz junctional EB (JEB-nH) demonstrated that genetic modifiers can contribute substantially to the phenotypic variability of JEB and likely other forms of EB. The innocuous changes in an 'EB related gene', *Col17a1*, have shown it to be a dominant modifier of *Lamc2^jeb^*. This work identifies six additional Quantitative Trait Loci (QTL) that modify disease in *Lamc2^jeb/jeb^* mice. Three QTL include other known 'EB related genes', with the strongest modifier effect mapping to a region including the epidermal hemi-desmosomal structural gene dystonin (*Dst-e/Bpag1-e*). Three other QTL map to intervals devoid of known EB-associated genes. Of these, one contains the nuclear receptor coactivator *Ppargc1a* as its primary candidate and the others contain related genes *Pparg* and *Igf1*, suggesting modifier pathways. These results, demonstrating the potent disease modifying effects of normally innocuous genetic variants, greatly expand the landscape of genetic modifiers of EB and therapeutic approaches that may be applied.

## Introduction

Epidermolysis Bullosa (EB) is a group of rare but often devastating genetic disorders characterized by fragility and blistering of skin and mucous membranes. These mechanobullous disorders are attributed to partial to full loss-of-function mutations in any of at least 20 genes whose protein products support the integrity of the epidermis and adhesions between the epidermal layers and the dermis (here termed 'EB related genes') [1, 2]. Lesions depend on the plane within the skin in which the gene product is normally localized. The EB simplex (EBS) subform characterized by cleavage within basal keratinocytes is attributed to mutations in *KRT5*, *KRT14*, *PLEC*, *DST*, *TGM5*, *PKP1*, *DSP*, *JUP*, *EXPH5* and *KLHL24* [1]; junctional EB (JEB) in which dermal-epidermal separation is within the basement membrane zone (BMZ) is caused

**Funding:** DCR received awards 'Roopenian 1' from Debra UK (https://www.debra.org.uk/) and 'Roopenian 2' from Debra Austria (https://www.debra-austria.org/). The Jackson Laboratory (jax.org) provided supplemental support for these studies. The funders had no role in study design, data collection and analysis, decision to publish, or preparation of the manuscript.

**Competing interests:** The authors have declared that no competing interests exist.

by mutations in *LAMA3*, *LAMB3*, *LAMC2*, *ITGA6*, *ITGB4*, *ITGA3* or *COL17A1*; and dystrophic EB (DEB), with separations in the dermis below the BMZ, is caused by mutations in *COL7A1*. Other identified subforms are caused by mutations in *FERMT1* and *CD151* [1, 3, 4].

As a general rule, all subforms of EB exhibit simple Mendelian patterns of inheritance but they can vary notably in presentation and severity [5]. While part of observed phenotypic differences is attributed to the nature of each primary mutation, substantial variance can be found among individuals carrying the same mutation [3, 6–8], potentially due to genetic or environmental modifier effects. Validation of genetic modifiers in human EB is challenging due to limited clinical information and the rarity of all subforms [9]. In one study *MMP1* was implicated as a modifier of DEB caused by a particular collagen VII recessive mutation [10] but two further studies with different *COL7A1* mutations failed to correlate *MMP1* alleles with disease variation in patients [11, 12]. A separate study of monozygotic twins homozygous for a recessive *COL7A1* DEB mutation found them to have differing levels of lymphotoxin α (LTA) in fibroblasts, implicating that as the cause of observed dramatic phenotypic differences between them. However, that study failed to determine whether genetic, epigenetic, environmental or other factors were responsible for the observed expression differences [13]. Formal proof of EB modifiers in humans remains to be established.

Mouse models provide a more tractable means to investigate the complex genetic basis of EB. The *Lamc2^{jeb}* mouse model carries a recessive mutant allele (*jeb*) of laminin gamma 2 (*Lamc2*) that encodes a subunit of the basement membrane extracellular matrix (ECM) molecule laminin 332 [14]. *Lamc2^{jeb/jeb}* homozygotes have insufficient levels of laminin 332 resulting in weakened dermal-epidermal adhesion, and consequently develop a syndrome that is a remarkable phenocopy of non-Herlitz JEB (JEB-nH) [15]. This model was used to provide the first strong evidence in support of the potent effects that background genetic variation can have on the JEB phenotype, proving that innocuous changes in the 'EB related gene' *Col17a1* serves a modifier of the *Lamc2^{jeb/jeb}* disease [16].

This study identifies multiple quantitative trait loci (QTL) in addition to *Col17a1* that determine the severity of JEB in the *Lamc2^{jeb/jeb}* mouse model. $F_2$ crosses matched for 'dominant' modifiers successfully revealed additional QTL. A B6 x 129X1 $F_2$-*Lamc2^{jeb/jeb}* cross, using strains that are naturally allelically matched for *Col17a1* (Mouse Phenome Database (MPD), Phenome.jax.org) [17] but phenotypically disparate [16], revealed 3 new QTL, including a 'dominant' modifier on chr1. Analysis of a MRL x FVB $F_2$-*Lamc2^{jeb/jeb}*, *Col17a1^{FVB/FVB}* cross (using a MRL.FVB-*Lamc2^{jeb/jeb} Col17a1^{FVB/FVB}* congenic strain) that was allelically matched for the newly identified chr1 dominant modifier (per MPD) uncovered 3 additional QTL. Though these QTL are 'dominated' by *Col17a1* and the chr1 QTL in $F_2$ crosses using tail tension tests as a quantitative measurement, the pathological phenotypes of ear and tail lesion severity support their contributions to disease. Finally, haplotype analyses, consideration of locations of 'EB related genes' and in some cases congenic strain confirmation and fine mapping by congenic reduction suggest strong candidates for these new QTL.

## Results

### Relative contribution of *Col17a1* versus other genetic factors to JEB disease severity in *Lamc2^{jeb/jeb}* mice

Strain comparisons of B6 vs.129, MRL vs. FVB and B6 vs. FVB, all homozygous *Lamc2^{jeb/jeb}*, have been shown to differ substantially in their severity of JEB [16]. To determine the relative contributions of *Col17a1* and other genetic components of mice genetically sensitized to JEB owing to homozygosity of the *Lamc2* mutation, *Lamc2^{jeb/jeb}* mice bred to be matched or disparate for *Col17a1* alleles were compared by phenotyping methods designed to reveal

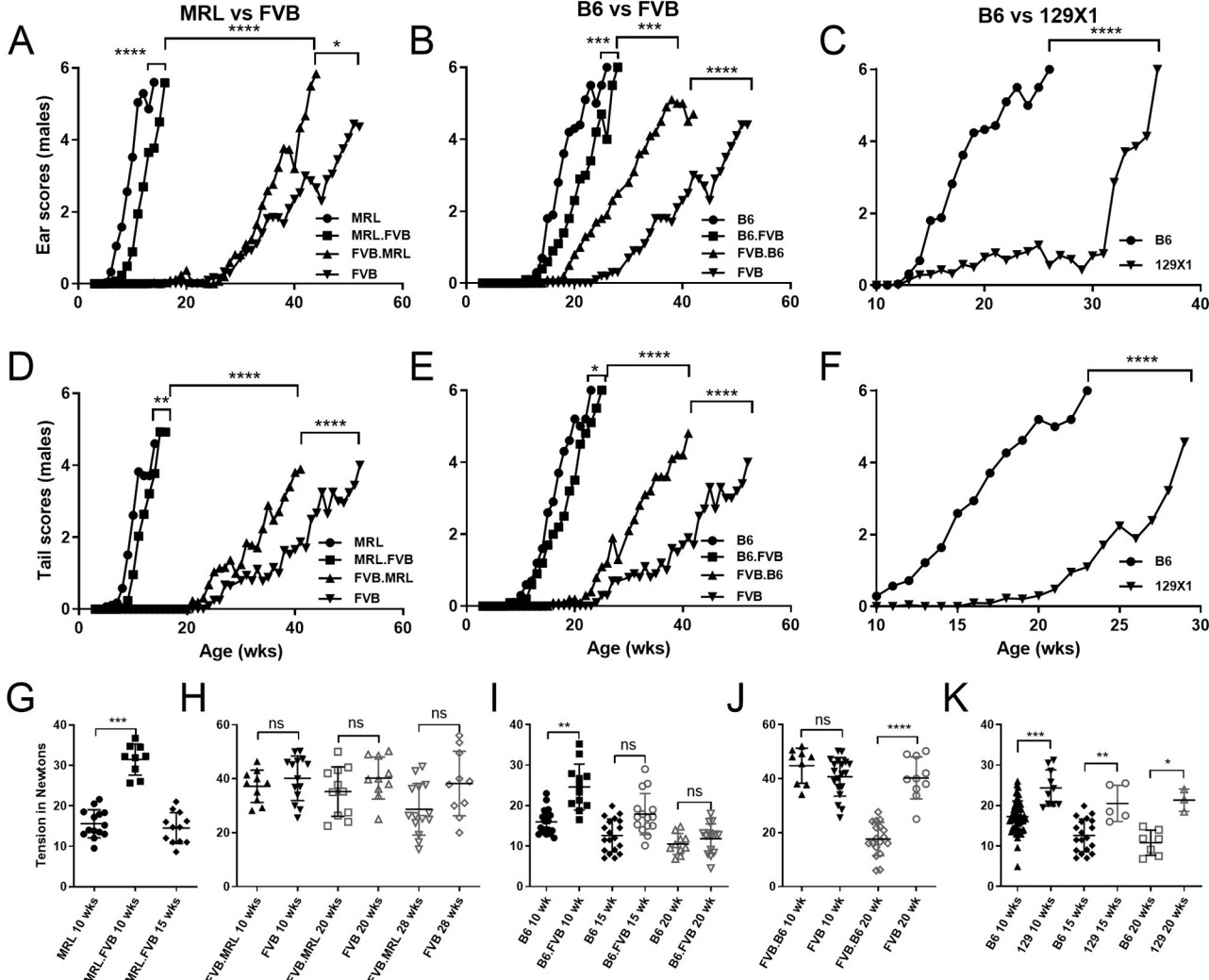

**Fig 1. Contributions of *Col17a1* and other genetics to *Lamc2^{jeb/jeb}* modifier effects.** (A-C) ear scores weekly, from 0 ('unaffected') to 6 ('very affected'). (D-F) tail scores using same method. (G-K) tension at various ages. All mice tested are male, *Lamc2^{jeb/jeb}* homozygotes. Congenic names are for *Col17a1* (*e.g.*, MRL.FVB is MRL-*Lamc2^{jeb/jeb} Col17a1^{FVB/FVB}*). Ear and tail score statistics are based on age when scores first reach '4' ('moderately affected') calculated as survival in PRISM by both Log-rank (Mantel-Cox) and Gehan-Breslow-Wilcoxon, with the less significant of the two indicated. G, MRL required euthanization before 15 wks due to their severity of JEB. Tension statistics are PRISM 1-way ANOVA. Tension bars indicate mean with SD. ns = not significant, * <0.05, ** <0.01, *** <0.001, **** <0.0001. Includes some B6, 129, FVB and MRL data previously published [16].

pathophysiological manifestations of JEB [16]. This includes the "tail tension test", which measures the force (in Newtons) required to separate the epidermal layer, and visual longitudinal scoring of the development and severity of ear and tail lesions (Fig 1). In all cases allelic variation in *Col17a1* had an appreciable effect but was still far from accounting for the full phenotypic differences among the strains, implicating additional genetic modifiers. Direct comparison of FVB to FVB.B6 and FVB.MRL *Col17a1* congenics (all *Lamc2^{jeb/jeb}*), which both differ from FVB at COLXVII AA 1277 while FVB.B6 also differs at COLXVII AA 1292, suggests that both AA impact binding strength and disease onset (Fig 2) [16]. Tension tests suggest AA 1292 has more impact on attachment strength (B6 differs from MRL and FVB, Fig 2A), while tail scores and survival data indicate AA 1277 has more impact on those clinical manifestations (B6 and MRL differ somewhat from each other, but both differ substantially

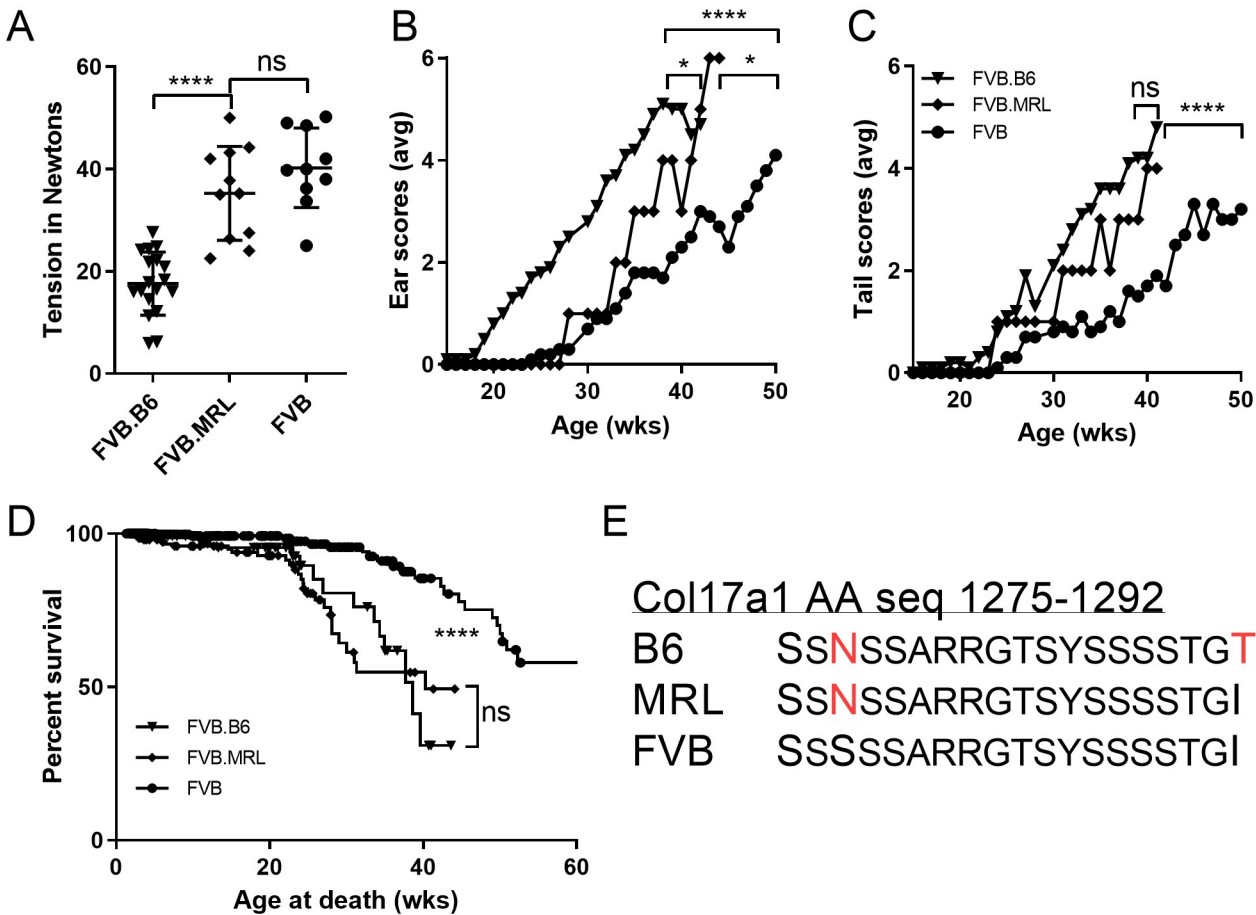

**Fig 2. Direct comparisons of B6 and MRL *Col17a1* congenics on FVB background.** (A) Tail tension test at 20 weeks of age. (B-C) Average ear and tail scores from 0 'not affected' to 6 'very affected' for mice scored weekly. (D) Cumulative censored survival data. (E) Amino acid sequence for *Col17a1* AA 1275–1292 for tested inbred strains, from Sproule et al 2014. All (A-D) data shown is for FVB males homozygous *Lamc2^{jeb/jeb}* and for B6 or MRL *Col17a1* congenic segments as indicated. (E) MRL differs from FVB at *Col17a1* AA 1277. B6 differs from FVB at both AA 1277 and 1292. Tension bars indicate mean with SD. Tension p-values are 1-way ANOVA. Ear and tail score p-values are less significant of Log-rank (Mantel-Cox) or Gehan-Breslow-Wilcoxon, based on age when each mouse first reaches a score of '4' treated as survival.

from FVB, Fig 2C and 2D). Ear scores suggest both contribute equally (Fig 2B). While FVB-*Lamc2^{jeb/jeb}* is the 'most resistant' strain background documented so far by tension tests, ear scores and tail scores, they have been noted to be more subject to sudden unexplained deaths, perhaps due to disease manifestations in other organs in which *Lamc2* is highly expressed, such as lungs, stomach and large intestines (biogps.org). Replacing the *Col17a1* allele with B6 or MRL exacerbates this phenotype (Fig 2D).

### Mapping of B6 vs 129X1 QTL that alter JEB phenotypes in *Lamc2^{jeb/jeb}* mice

To further address the possibility of genetic modifiers, a cohort of 268 male B6 x 129X1 $F_2$-*Lamc2^{jeb/jeb}* homozygotes, naturally matched for *Col17a1*, along with parental B6-*Lamc2^{jeb/jeb}*, 129X1-*Lamc2^{jeb/jeb}* and $F_1$-*Lamc2^{jeb/jeb}* controls were evaluated at 10 weeks of age by the tail tension test and values in Newtons were recorded (Fig 3A). $F_2$ mice were genotyped for simple sequence length polymorphisms (SSLP) markers across the genome and genome wide

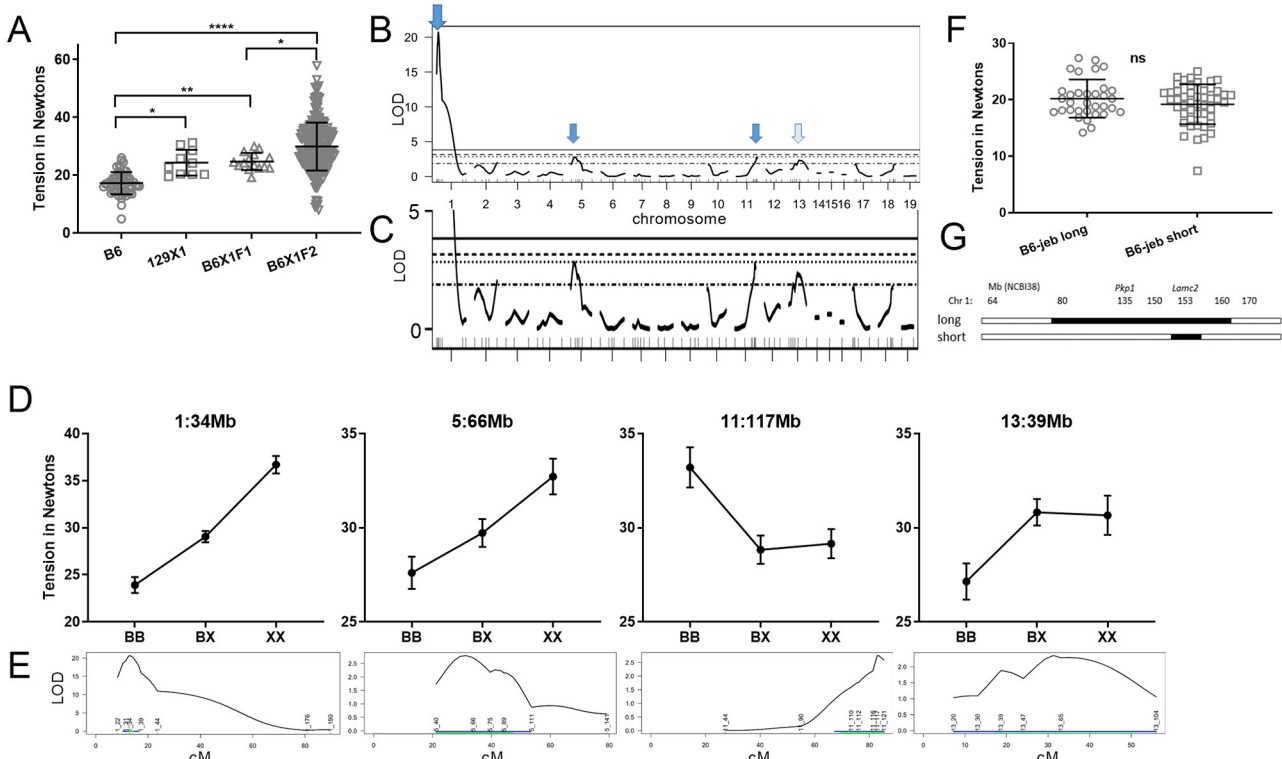

**Fig 3. (B6 x 129X1)F$_2$-*Lamc2^{jeb/jeb}* tail tension R/QTL.** (A) Tension test results for parental B6 and 129X1, F1 and F2 male mice at 10 weeks of age (P-values: * < 0.05, ** <0.01, **** <0.0001). (B) LOD score of tension x marker analysis for the B6X1F2 tension set with LOD score of 0–5 expanded in (C) to show greater detail. Horizontal lines bottom to top are 63, 10, 5 and 1% cutoffs. (D) Effect plots for SSLP markers nearest the peaks. X-axis is genotype: BB = B6, BX = het, XX = 129X1. (E) 95% confidence intervals for the four most significant QTL. Positions of SSLP markers tested with Mb position names are indicated (i.e. 1_22 is chr1, 22Mb). Blue lines indicate 1.5 LOD drop interval. Green lines indicate Bayesian confidence intervals. 95% CIs converted to Mb are: chr1:22-39Mb (peak 34Mb), chr5:40-111Mb (peak 58Mb), chr11:90-122Mb (peak 119Mb) and chr13:20-104Mb (peak 59Mb). (F) Tail tension comparison of 'long' and 'short' B6.129X1-*Lamc2^{jeb/jeb}* congenics. (G) Mb map of same. 129 is black, B6 is white. All mice are male, *Lamc2^{jeb/jeb}* homozygotes. Tension bars indicate mean with SD.

associations were calculated using the R/QTL algorithm [18, 19]. The most significant value encompassed chr1:22-39Mb (95% confidence interval), peaking at 34Mb with a LOD score >20. Two additional QTL achieved genome wide significance of *P* <0.10 with 95% CIs of 5:40-111Mb (peak at 58Mb) and 11:90-122Mb (peak at 119Mb) (Fig 3B–3D). A fourth suggestive QTL at 13:20-104Mb (peak at 59Mb) that achieved minimal significance but proved to be relevant to studies described subsequently is also noted.

The *Lamc2^{jeb}* mutation arose in 129X1 mice [15]. The stock of B6-*Lamc2^{jeb/jeb}* mice used in the B6129F2 mapping cross carry >80Mb of adjoining 129X1 chromatin and thus would not reveal B6 vs 129X1 QTL which may map to that region. To address that region, tension comparisons were made of the >80Mb long congenic and a second <10Mb short congenic B6-*Lamc2^{jeb/jeb}* which was generated as part of this study (Fig 3E). Long and short congenic tension values did not statistically differ, indicating that B6/129 variants, including the 'EB related gene' *Pkp1*, within the disparate chr1 congenic segments (Fig 3F) do not significantly alter the JEB phenotype.

## Mapping of MRL.FVB-*Col17a1^{FVB/FVB}* vs FVB QTL

Similar to above, a cohort of 176 male mice were generated by an F$_2$ intercross of MRL.FVB-*Lamc2^{jeb/jeb}* *Col17a1^{FVB/FVB}* x FVB-*Lamc2^{jeb/jeb}* (homozygous *Lamc2^{jeb/jeb}* and matched

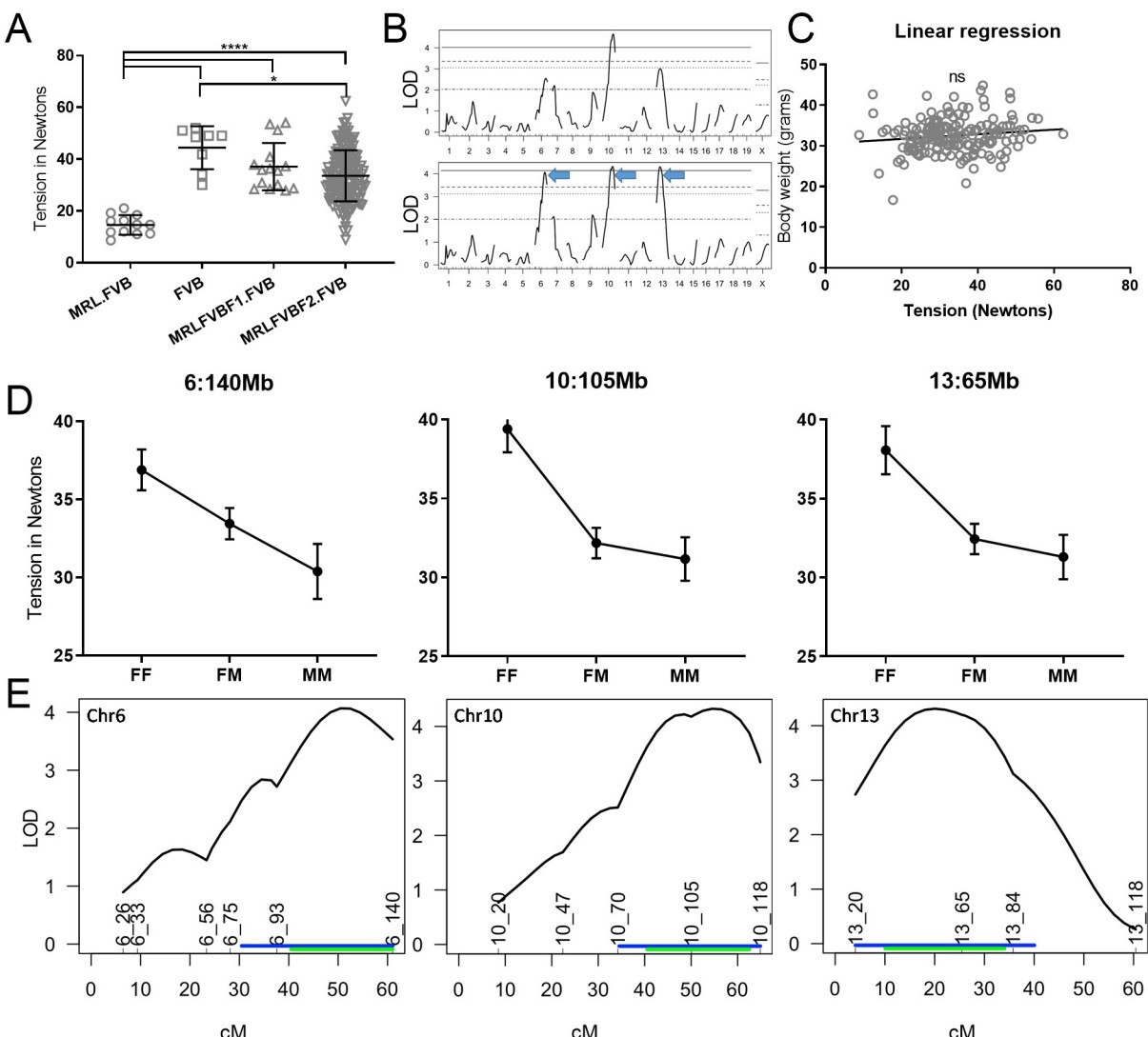

**Fig 4. (MRL.FVB x FVB)F₂-*Lamc2^jeb/jeb* *Col17a1^FVB/FVB* tail tension R/QTL.** (A) Male 15 week old tail tension test results for MRL.FVB x FVB F₂ (n = 176) with F₁ and parental FVB and MRL.FVB controls (all homozygous *Lamc2^jeb/jeb* and *Col17a1^FVB/FVB*). Bars indicate mean with SD. (B) R/QTL LOD scores showing peaks on chr6, 10 and 13. (C) Inclusion of body weight as a covariant increased significance of chr6 and 13 peaks. (D) Effect plots for SSLP markers nearest the QTL peaks. (E) 95% confidence intervals for the three most significant QTL. Converted to Mb are: 6:66-127Mb (peak 110Mb), 10:66-117Mb (peak 104Mb) and 13:11-74Mb (peak 41Mb). Positions of SSLP markers tested with Mb position names are indicated (i.e. 6_26 is chr6, 26Mb). Blue lines indicate 1.5 LOD drop interval. Green lines indicate Bayesian confidence intervals. (F) Linear regression of tension vs body weight for F₂ mice is not statistically significant.

homozygous for the FVB allele of *Col17a1*). They were genotyped for 89 MRL vs FVB disparate SSLP markers distributed throughout the genome (S1 Table) and tension tested at 15 weeks of age and compared to parental and F₁ controls (Fig 4A). We had noted that the MRL.FVB F2 mice varied appreciably in body size, potentially as a manifestation of systemic effects. We thus weighed all mice when euthanized and used such data as a covariant. R/QTL analysis identified major QTL at 6:66-127Mb (peak at 110Mb), 10:66-117Mb (peak at 104Mb) and 13:11-74Mb (peak at 41Mb), with the chr6 and 13 QTL both increased in significance when 15-week-old body weight was considered as a covariant (Fig 4B–4E). Linear regression showed correlation between body weight and tension was not statistically significant (p = .08) (Fig 4F).

## Cross-comparing LOD plots

Comparing the B6/129 and FVB/MRL LOD plots suggest chr13 as the only QTL common to both crosses, as it exceeds the 0.63 (but not the 0.10) threshold in the B6/129 cross. Lack of other peaks suggests that B6 and 129 are allele matched at the chr6 and 10 QTL while FVB and MRL are matched at chrs1, 5 and 11. The following genotype correlations at the various QTL are thus concluded: B6/129 and FVB = MRL for chr1, 5 and 11; FVB/MRL and B6 = 129 for chrs6 and 10; and 129 = FVB/B6 = MRL for chr13. A summary of observed modifier QTL and effects for B6, 129X1, FVB and MRL across the observed QTL is included in Table 1. The Mouse Phenome Database (MPD) CGD-MDA1 SNP set was used to map the percent polymorphism by 0.1Mb increments across each QTL to discern patterns of strain matched or disparate heritage. Because the vast majority of mutations in mice predate the relatively modern development of inbred strains, the polymorphisms responsible for observed modifier effects are more likely to be historic [20], and to map to regions of disparate heritage between strain pairs which identify QTL and regions of shared heritage between strains predicted to be allele matched at each QTL. These results are discussed separately for each chr/QTL below.

## Chr1/*Dst-e*

Genotypes were predicted to be B6/129 and FVB = MRL for the chr1 QTL. The MPD CGD-MDA1 percent polymorphism map comparing B6 and 129X1 reveals them to have disparate heritage across ~1/2 of the 17Mb QTL, reducing the 'likely' candidate interval. Limiting to regions where FVB = MRL does not substantially reduce this candidate interval (Fig 5A–5C). The "EB related" gene, dystonin (*Dst/Bpag1*) [21, 22] maps to this interval. Dystonin has three main isoforms expressed predominantly in nerves (*Dst-a*), muscle (*Dst-b*) and epithelial tissue (*Dst-e*) [23]. *Dst-e* has a known role in EB, prominent epidermal expression (biogps.org) and favorable position within the QTL (very near the QTL peak and mapping within a predicted sub-region (B6/129 and FVB = MRL, Fig 5C)), all of which make it an attractive modifier candidate. While the 17Mb QTL contains 282 other genes (79 named genes, 180 'Gm' and 23 'Rik'), none of them other than *Dst-e* are obvious modifier candidates.

To further assess this QTL, we took advantage of available B6-background mouse stocks carrying congenic or consomic chr1 chromatin. The B6.129X1-*Zap70^{tm1Weis}*/J congenic strain was crossed to B6-*Lamc2^{jeb/jeb}* mice to produce B6-*Lamc2^{jeb/jeb}* 1:22-37Mb^{129/129}congenics. (These mice are also *Zap70^{-/-}*, but that gene is unlikely to be involved in an EB pathway.)

**Table 1. Phenotypes of QTLs and percent of variation explained.**

| Strain | 1/*Dst* | 5/*Ppargc1a* | 6/*Pparg* | 10/*Igf1* | 11/*Itgb4* | 13/*Dsp* | 19/*Col17a1*\* | Sum |
|---|---|---|---|---|---|---|---|---|
| | \multicolumn{8}{c}{**Chr/suggested, suspected or known (\*) gene**} | | | | | | | |
| 129X1 | R | R | = | = | S | R | S | R |
| B6 | S | S | = | = | R | S | S | S |
| FVB | S | = | R | R | = | R | R | R |
| MRL | S | = | S | S | = | S | S | S |
| | \multicolumn{8}{c}{**Percent of tension variation explained**} | | | | | | | |
| B6 vs 129X1 | 38 | 18 | 0 | 0 | 19 | 14 | 0 | |
| FVB vs MRL.FVB | 0 | 0 | 21 | 26 | 0 | 22 | 6\*\* | |

R = resistant (higher tail tension, later onset), S = susceptible (lower tail tension, earlier onset). = indicates two strains phenotypically matched for a QTL but no not know if both resistant or both susceptible.

\*\*from previously published combined B6xFVB and 129X1xDBA1 $F_2$ crosses.

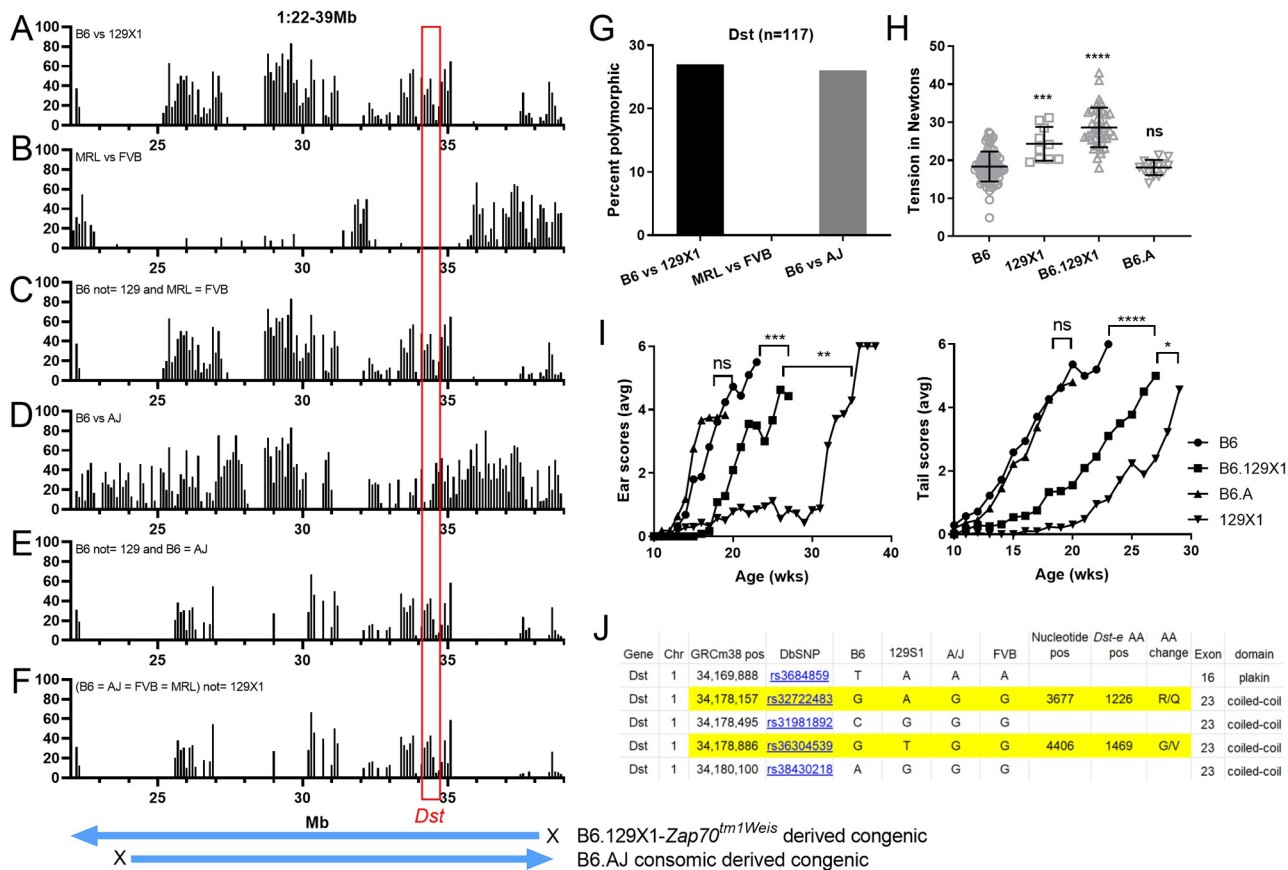

**Fig 5. Chr1 QTL analysis.** (A-F) Mouse Phenome Database (MPD) CGD-MDA1 percent polymorphism plots (PPP) across 1:22-39Mb QTL 95% confidence interval: (A) B6 vs 129X1 percent polymorphic, (B) FVB/MRL and (C) B6/129X1 polymorphic while FVB/MRL matched. Each peak/ bar = 0.1 Mb. Map beneath indicates extent of congenic segments tested. Red box indicates *Dst* position. (G) MPD PPP for *Dst*. (H) B6 chr1 congenic 10 wk old male tail tension test. All homozygous *Lamc2^{jeb/jeb}*. B6.129 and B6.AJ homozygous for chr1 congenic intervals 1:22-37Mb and 1:25-44Mb respectively. Bar indicates mean with SD. (D) Male ear and tail scores for the same congenics. (J) B6/129 missense SNPs mapping to *Dst-e*. A/J matches B6 at 2 of 5. Statistics in (H) are PRISM 1-way ANOVA all compared to B6-*Lamc2^{jeb/jeb}* control (B6) except as indicated. Statistics in (I): both Log-rank (Mantel-Cox) or Gehan-Breslow-Wilcoxon tests were performed, based on age when each mouse first reaches a score of '4' treated as survival. Values shown are the less significant of the two. \*\*\*\* is <0.0001, \*\*\* is <0.001, \*\* is <0.01, \* is <0.05, ns is not significant.

Because *Dst* is a prime candidate for the chr1 modifier and A/J is allelically intermediate between B6 and 129 at *Dst* (Fig 5E **and MPD**), it can be used to reduce a candidate gene polymorphism list. The C57BL/6J-Chr 1^{A/J}/NaJ consomic was used to produce B6-*Lamc2^{jeb/jeb}* 1:25-44Mb^{AJ/AJ}. Males from both were tail tension tested and ear and tail scored at 10 to 40 weeks of age. 129 congenics had higher tension values and later onset than B6-*Lamc2^{jeb/jeb}* controls, confirming the QTL (Fig 5H and 5I). A/J congenics did not, indicating that A/J should share the B6 allele rather than the 129 allele at the responsible genetic modifier locus (129/ B6 = A/J) unless the modifier maps to the proximal part of the QTL, outside the A/J congenic (22-25Mb) interval. This is unlikely since it is a region of B6/129X1 shared strain heritage (Fig 5A). B6-*Lamc2^{jeb/jeb}* 1:22-37Mb^{129/129} were crossed to a B6-*Lamc2^{jeb/jeb}* Col17a1^{PWD/PWD} congenic (R03Q) [16] to produce and test a B6-*Lamc2^{jeb/jeb}* strain homozygous for both protective congenic intervals. In addition to the expected increase in 10-week male tail tension values, the skin was very resistant to removal, indicating that allelism at these two loci (*Col17a1* and the *Dst*-linked *QTL*) can go a long way toward determining susceptibility or resistance to disease caused by primary genetic defects of JEB (Fig 6).

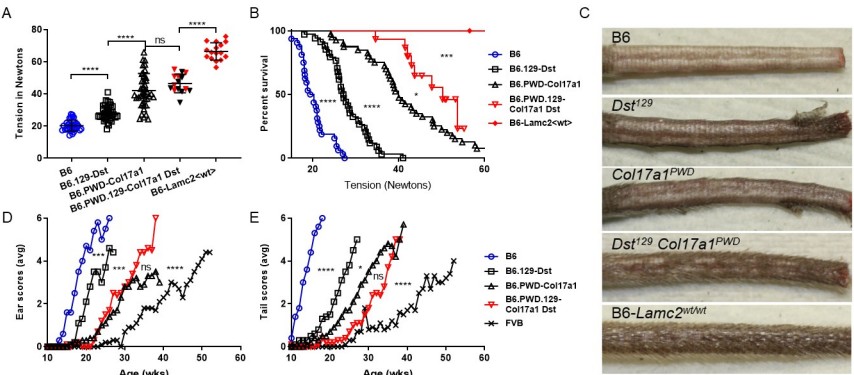

**Fig 6. *Col17a1* and 1/22-37Mb (*Dst*) double congenic effects on the *Lamc2<sup>jeb/jeb</sup>* disease.** (A) 10-week-old male tension comparisons of B6, B6-*Dst<sup>129/129</sup>* congenic, R03Q B6-*Col17a1<sup>PWD/PWD</sup>* congenic, B6-*Col17a1<sup>PWD/PWD</sup> Dst<sup>129/129</sup>* double congenic and B6 wild-type (B6-*Lamc2<sup>wt/wt</sup>*) (all *Lamc2<sup>jeb/jeb</sup>* except the last). Red symbols indicate clamp slipped off without removing substantial skin. (B) Tension expressed as survival of same with red values censored gives statistical difference between B6-*Col17a1<sup>PWD/PWD</sup>* and double congenic. (C) Photos showing the typical amount of skin removed during the tail tension test from (top to bottom): B6-*Lamc2<sup>jeb/jeb</sup>*, B6-*Lamc2<sup>jeb/jeb</sup> Dst<sup>129/129</sup>*, B6-*Lamc2<sup>jeb/jeb</sup> Col17a1<sup>PWD/PWD</sup>*, B6-*Lamc2<sup>jeb/jeb</sup> Dst<sup>129/129</sup> Col17a1<sup>PWD/PWD</sup>* double congenic and B6 wild-type, all at 10 wks of age. (D-E) Ear and tail scores for the same excluding B6 wt control and adding FVB-*Lamc2<sup>jeb/jeb</sup>* for comparison. All males. All *Lamc2<sup>jeb/jeb</sup>* except 'B6-*Lamc2<sup>wt/wt</sup>*'. *Dst* is contained within the 1/22-37Mb congenic and used here to represent it. Ear and tail statistics are survival based on age when each mouse first received a score of '4' (moderately affected). $^{****}$ p < .0001, $^{***}$ p < .001, $^{**}$ p < .01, $^{*}$ p < .05, ns = not significant.

Since the epithelial form of *Dst* (*Dst-e* or *Bpag1-e*, Ensembl build 38 mouse transcript Dst-213 [23]) is the most obvious QTL candidate in the chr1 interval, it was further evaluated to consider possible responsible polymorphisms. Per MPD, 129X1 and 129S1 are allele matched throughout *Dst* and the bracketing region (Fig 7), so 129S1 could be used to represent 129X1 in a Sanger Mouse Genome Project (REL-1410) polymorphism survey. Sanger's default search, which excludes intronic, intergenic and unspecified SNPs, documents 231 SNPs and 32 indels between B6 and 129S1 in the region 1: 34,160,330–34,183,904 which corresponds to the *Dst-e* transcript of dystonin (DST-213, per GRCm38.p6). Further excluding upstream, downstream, Nc transcript, non-coding exon, synonymous, splice region and 3'UTR variants as unlikely to be responsible for the modifier phenotype reduces the list to only 5 SNPs, all missense. Results from the B6.AJ congenics further reduce the candidate missense list to only the two at which B6 and A/J have the same allele: R1226Q and G1469V (Fig 5J).

## Chr5/*Ppargc1a*

The chr5 MPD percent polymorphism maps reveal large regions of B6/129 shared heritage in the proximal portion of the QTL, while the distal half contains much more disparity. Similar to chr1 results, factoring in regions where FVB and MRL are allele matched again does little to reduce the 'likely' QTL interval (Fig 8A). No known EB related genes map to the chr5 QTL.

Three B6.129 strains (B6N.129S4-*Ppargc1a<sup>tm2.1Brsp</sup>*/J, B6.129P2-*Sod3<sup>tm1Mrkl</sup>*/J, B6.129S4 (Cg)-*Dck<sup>tm1.2Radu</sup>*/J) whose congenic intervals are included in the chr5 QTL were crossed to B6-*Lamc2<sup>jeb/jeb</sup>* to produce double homozygotes and phenotyped to address the majority of the 5:40-111Mb QTL (covers 5:41-93Mb). Select chr5 recombinants produced during breeding were also bred to homozygosity and tested to further reduce the candidate interval.

Male mice were tension tested at 10 weeks of age. Overlapping congenics R01, R04, R21 and R22 confirmed a 129 allele-resistance modifier locus on chr5 at ~51Mb (Fig 8B). Other partially overlapping non-protective congenics (R03, R32, R33), along with Sanger and MPD

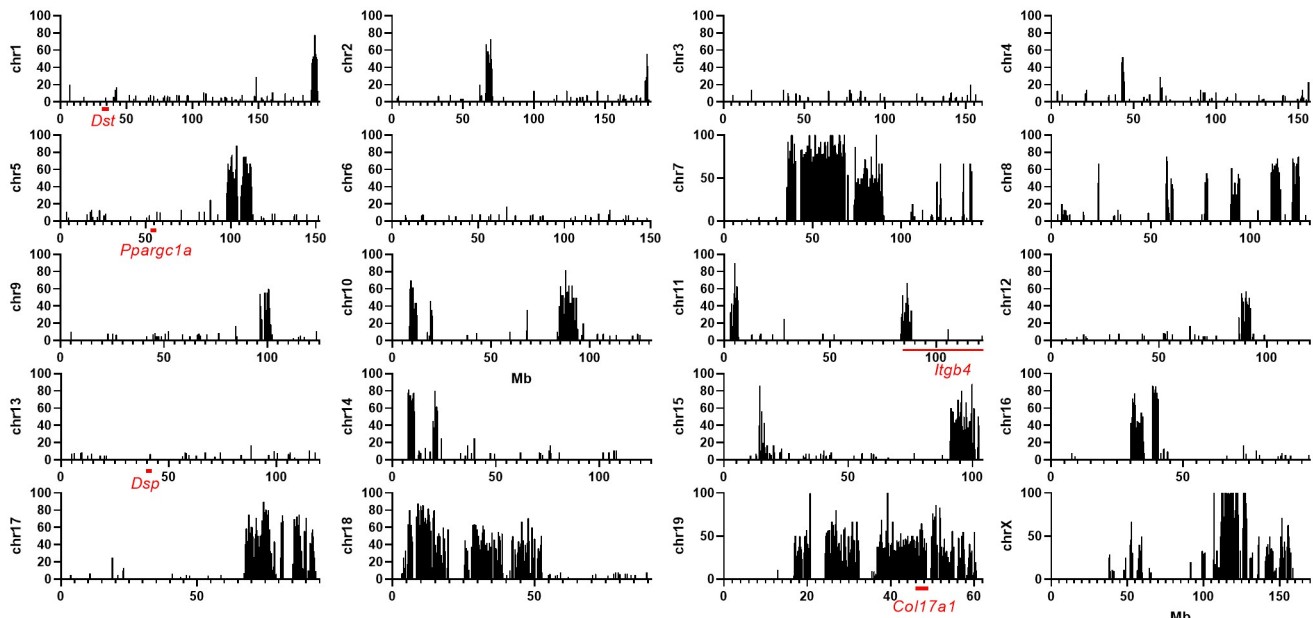

**Fig 7. Polymorphism plots.** Mouse Phenome Database CGD-MDA1 percent polymorphism plots of 129X1/SvJ (129X1) vs 129S1/SvImJ (129S1) throughout the genome by 0.1Mb increments, limited to SNPs with allele data for both strains. B6 x 129X1 F2s give QTL. B6 vs 129S1, but not 129X1, full genome SNP polymorphism data is available in Sanger. Percent polymorphism plots reveal regions of the genome where 129S1 can fairly be used as a substitute for 129X1 in Sanger B6 vs 129 SNP analysis, including all regions of interest to us in this study: 1:22-39Mb/*Dst*, 5:50-53Mb/*Ppargc1a*, 11:90-122Mb/*Itgb4* and 13:38-39/*Dsp* (red underlines). *Col17a1* on chr19, which *does* genetically differ between 129X1 and 129S1, is also underlined for reference.

information showing a large region of B6 and 129 shared strain heritage at 5:45.3–50.5Mb (Fig 8A) restricted our candidate interval to 5:50.5–52.366Mb (Fig 8C).

The 5:50.5–52.366Mb candidate interval is relatively small and contains only 3 named genes (*Ppargc1a*, *Dhx15* and the start of *Sod3*), along with 23 predicted 'Gm' genes, 2 'Rik' genes and *Mir6417*. Per a standard Sanger search using default values (REL-1505, excludes intronic, intergenic and unspecified SNPs), the reduced candidate interval includes 408 SNPs, 125 indels and 24 structural variants between B6 and 129S1 (which can again be used to represent 129X1 due to shared heritage in the region, per MPD (Fig 7)), of which the vast majority map to *Ppargc1a*. However, 288 SNPs are listed as NMD transcript variants within *Ppargc1a* which are found to all map to introns. Removing them from consideration, along with upstream and downstream variants which are unlikely to affect phenotype leaves only 7 SNPs (4 3'UTR, 1 missense, 1 synonymous and 1 splice region variant, all in *Ppargc1a*), 6 indels and 24 structural variants in the entire interval.

*Dhx15* lists only 1 upstream SNP and 2 indels. And while *Sod3* includes 13 SNPs (2 missense, 3 synonymous and 8 5'UTR) and 2 indels, they are all excluded from the candidate interval by recombinant break points. The preponderance of evidence suggests a *Ppargc1a* polymorphism as most likely responsible for the observed phenotypic difference and QTL (Fig 8F).

Rechecking MPD SNPs for the 5:50.5–52.366Mb reveals a high B6/129 polymorphism percentage with very low FVB/MRL percentage, particularly in candidate gene *Ppargc1a* (Fig 8E), which matches the prediction based on the absence of a R/QTL FVB vs MRL LOD peak.

### Chr11/*Itgb4*

The 11:90-122Mb (peak at 117Mb) QTL contains EB related gene *Itgb4*, mapping ~3Mb from the QTL peak. Other EB related genes mapping within the 95% confidence interval but further

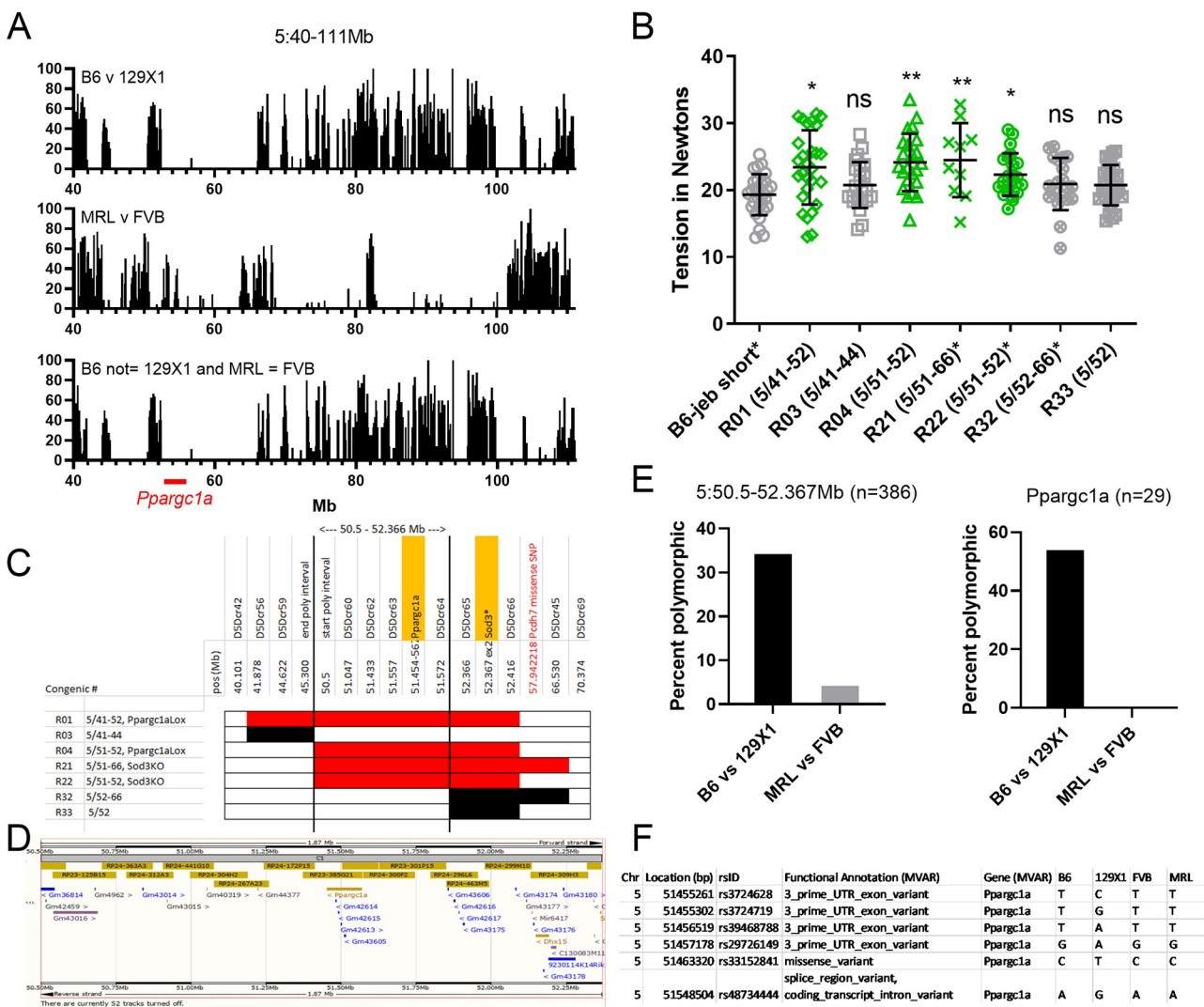

**Fig 8. Chr5 QTL analysis.** (A) Percent MPD CGD-MDA1 SNP polymorphism map (each bar = 0.1 Mb) for the chr5 QTL interval. Clustered bars indicate regions of disparate strain heritage. Comparison of B6/129 (top, differ at QTL) to FVB/MRL (middle, predicted to match at QTL) does little to reduce candidate interval (bottom, B6/129 and FVB = MRL). (B) Chr5 congenic 10 wk old male tail tension test. * after name indicates cross made using a reduced B6-*Lamc2jeb/jeb* congenic (<10 Mb) instead of originally published (>80 Mb). Statistical significance for each congenic is based on comparison to its parental short or long congenic strain. (C) Focused chr5 congenic map of protective (red) and non-protective (black) lines limits the main candidate interval to 50.5–52.366 Mb. (D) Ensembl view of 5:50.5–52.366Mb showing genes present. (E) Percent polymorphism plots for reduced congenic interval and candidate gene *Ppargc1a* from the MPD CGD-MDA1 SNP data set. (F) List of B6 vs 129X1 polymorphisms in *Ppargc1a* excluding synonymous and intronic from GenomeMUSter GRCm38 (mpd.jax.org/genotypes). All are predicted MRL = FVB.

from the peak are *Itga3*, *Krt14* and *Jup* at 94.9, 100.0 and 100.2Mb respectively. *Itga3*, *Krt14* and *Jup* all map to regions which do not include the predicted genotype combination B6/129 and MRL = FVB (Fig 9A). A B6-*Lamc2jeb/jeb* 11:110-118Mb$^{129/129}$ congenic was made using B6.129S1-*Stat3tm1Xyfu*/J and tension tested, but values did not statistically differ from B6-*Lamc2jeb/jeb* controls (Fig 9C), failing to confirm the QTL. This may be due to the weak nature of the QTL or that the modifier locus maps outside the congenic interval and does not necessarily disprove the QTL. The closest markers tested and not in the 11:110-118Mb congenic were at 90 and 121Mb.

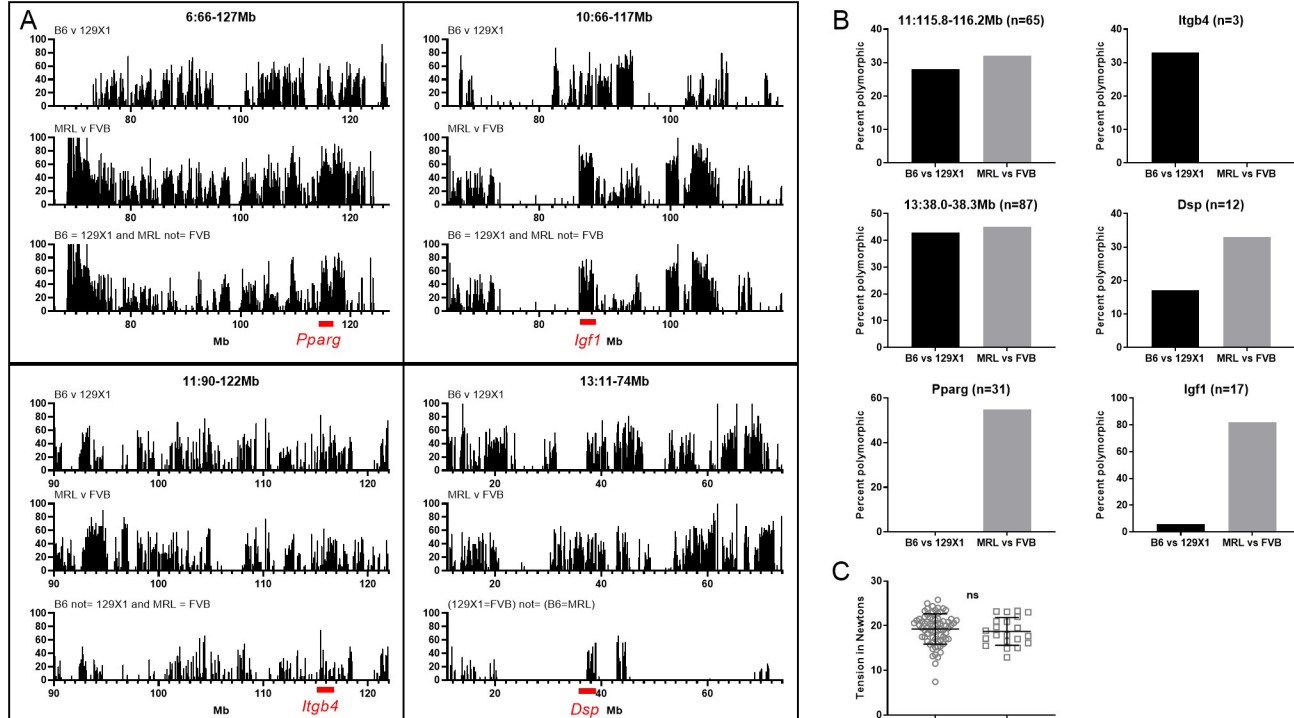

**Fig 9. Remaining QTL analyses.** (A) Strain heritage comparisons across QTLs: B6 v 129 top, MRL v FVB middle and appropriate prediction-based combined plot bottom for each. Bars indicate percent polymorphism among MPD CGD-MDA1 SNP markers in 0.1Mb increments, only looking at markers which contained genotype information for all 4 strains. Labels indicate full congenic intervals. Red underline indicates location of genes of interest. (B) MPD percent polymorphism for strain pairs for specific candidate intervals and genes. (C) Tail tension test of 10 week old male chr11 QTL congenic vs B6 control. Both groups *Lamc2ʲᵉᵇ/ʲᵉᵇ*. Bars indicate mean with SD.

Data mining of Sanger was performed to consider candidate *Itgb4* polymorphisms which could be responsible for the observed phenotypic differences. MPD CGD-MDA1 11:100-121Mb reveals 129X1/129S1 shared strain heritage (3865 129S1/129X1 SNPs with only 1 polymorphic), so 129S1 can be used as a proxy for 129X1 in Sanger *Itgb4* and chr11 QTL region queries (Fig 7). A default Sanger REL-1505 *Itgb4* query for 129S1 reveals 65 B6/129 SNP polymorphisms (excludes intronic, intergenic and unspecified), 20 insertions/deletions and 1 structural variant. The SNPs are mostly upstream, downstream and synonymous, but include

**Table 2. Candidate SNPs in *Itgb4* and *Dsp*.**

| Chr | Location (bp) | rsID | Functional Annotation (MVAR) | Gene (MVAR) | B6 | 129X1 | FVB | MRL |
|---|---|---|---|---|---|---|---|---|
| 11 | 115991899 | rs264509461 | missense_variant | *Itgb4* | A | G | G | G |
| 11 | 115997947 | rs29430524 | splice_region_variant, coding_transcript_intron_variant | *Itgb4* | T | C | C | C |
| 11 | 116000680 | rs27001410 | splice_region_variant, synonymous_variant | *Itgb4* | C | A | A | A |
| 11 | 116006569 | rs27001391 | missense_variant | *Itgb4* | T | C | C | C |
| 13 | 38191736 | rs29528874 | missense_variant | *Dsp* | G | A | G | A |
| 13 | 38196709 | rs387340276 | missense_variant | *Dsp* | T | C | T | C |

SNPs in *Itgb4* with genotype pattern B6/129X1 and MRL = FVB and SNPs in *Dsp* with genotype pattern 129X1 = FVB/B6 = MRL excluding intronic and synonymous from GenomeMUSter GRCm38 (mpd.jax.org/genotypes).

3 missense and 2 splice region variants. Candidate *Itgb4* SNPs identified as missense or splice region are listed in Table 2.

### Chr13/*Dsp*

The chr13 QTL is the only one implicated in both crosses. It is allelically predicted to be 129 = FVB / B6 = MRL at the QTL. Application of this information to the MPD polymorphism map reduces the 'likely' candidate intervals to only a few punctate regions, totaling not more than 6Mb (Fig 9A). *Dsp* (desmoplakin) is the sole known EB related gene mapping to the 63Mb QTL, and it also maps to the <6Mb reduced interval, ~3Mb from the QTL peak. MPD check of the 13:38–38.3Mb interval bracketing *Dsp* indicates 129X1 and 129S1 are matched (Fig 7), so 129S1 can be used to represent 129X1 for *Dsp* in Sanger. Per Sanger, B6 differs from 129 at 35 SNPs in *Dsp* (1 missense, 11 synonymous, 16 upstream and 8 downstream gene variants) and 11 indels, of which 129 = FVB (predicted genotype) at 25 SNPs and 6 indels, including the sole B6/129 missense SNP. Since MRL is not in Sanger, this candidate list could not be limited further without sequencing, which was not done. Candidate *Dsp* missense SNPs with the predicted genotype pattern are listed in Table 2.

### Chr6 and 10 strain heritage analyses

MPD percent polymorphism maps for the chr6 and 10 QTL, each predicted to allelically be 'FVB/MRL and B6 = 129' at the modifier locus have reduced 'likely' candidate intervals which are ~1/2 to 3/4 of the original QTL (Fig 9A) and contain many genes (not listed), though no known EB related genes. Notably, they do each contain a gene that interacts with or is related to chr5 candidate *Ppargc1c*: *Pparg* on chr6 [24] and *Igf1* on chr10 [25, 26]. However, a GenomeMUSter survey of SNPs for *Pparg* and *Igf1* did not find good candidates such as missense mutations that matched the predicted genotype pattern of FVB / MRL and B6 = 129X1. Therefore, no chr6 or chr10 genetic modifier candidates are offered in Table 2 to accompany those from chr11 and ch13.

### Relative impact of modifiers

Cross comparison of ear and tails scores and tail tension test for the four strains used in these mapping crosses reveal more phenotype 'spread' between FVB and MRL than between B6 and 129X1, even when FVB and MRL are matched for *Col17a1* (Fig 10), indicating that the modifiers responsible for the FVB/MRL differences have greater effect than those operating in the B6/129X1 comparison.

## Discussion

Epidermolysis bullosa had long been considered a simple Mendelian inherited genetic disorder due to correlation between presence of disease and single gene primary mutations [27], though the possibility of genetic modifiers has been increasing recognized [3, 9]. Disease severity varies widely among EB patients suffering from mutations in the same gene, but the rarity of specific alleles has made it difficult to determine in humans the extent to which symptoms were related to the primary disease mutation and to what extent they may be modified by other genetic or environmental factors [5, 28]. Titeux et al. correlated alleles of the protease *MMP1* with dystrophic EB disease severity in individuals homozygous for a hypomorphic allele of *COL7A1*, apparently because differing rates of MMP1 degradation of collagen VII resulted in critical differences in the amount of collagen VII available for binding to laminin 332 at the dermal-epidermal boundary [10]. This shares similarities with these mouse studies

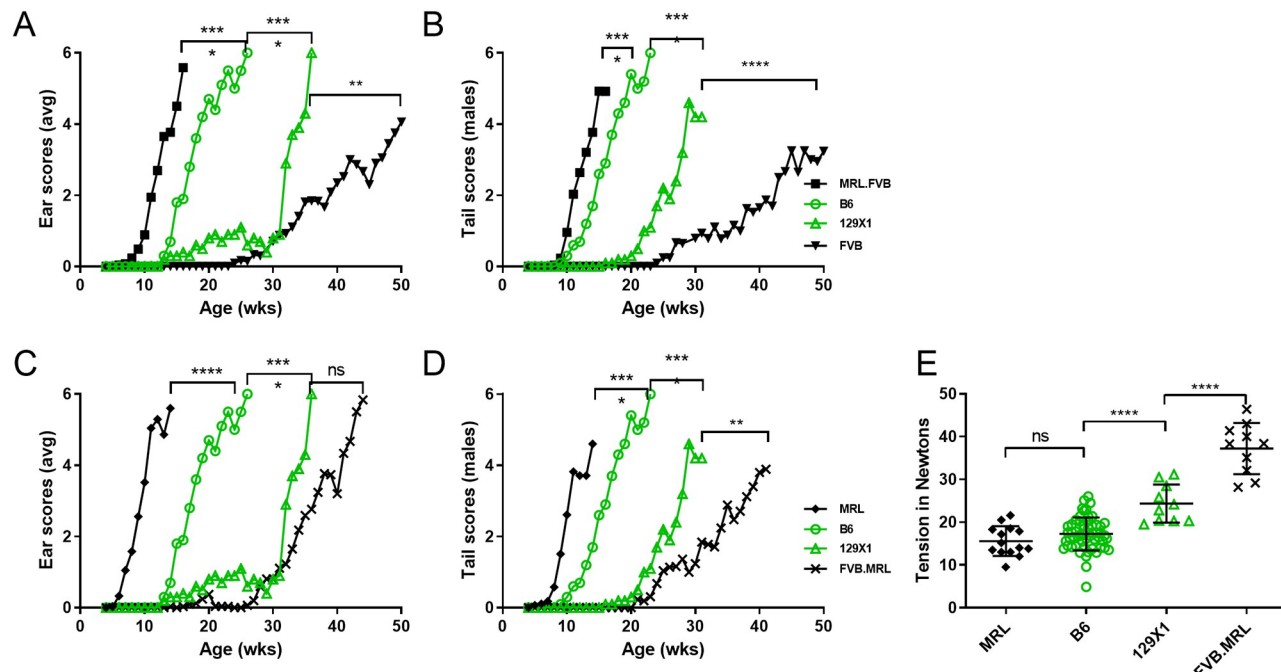

**Fig 10. Comparative scoring.** Comparison of ear (A, C) and tail (B, D) scores for *Col17a1* allele matched B6 and 129X1 versus MRL and FVB matched for *Col17a1^{FVB/FVB}* (A, B) or *Col17a1^{MRL/MRL}* (C, D). All scores are from 0 ('unaffected') to 6 ('very affected'). (E) Tension comparison of B6, 129, MRL and FVB 10 week old males where B6 and 129 are naturally *Col17a1*-matched and MRL and FVB are congenically matched for *Col17a1^{MRL/MRL}*. All (A-E) mice are male, homozygous *Lamc2^{jeb/jeb}* and homozygous for the indicated alleles of *Col17a1*. Ear and tail score statistics (A-D) are less significant of Log-rank (Mantel-Cox) or Gehan-Breslow-Wilcoxon based on age each mouse first to score '4'. Tension statistics (E) are 1-way ANOVA. Tension bars indicate mean with SD.

in which homozygosity for the hypomorphic *Lamc2^{jeb}* allele results in minimal production of 'healthy' laminin 332 and sensitivity to modifier effects of other genes. Using this mouse model, it was shown that *Col17a1* non-collagenous domain 4 (NC4) missense polymorphisms (innocuous in the context of healthy alleles of other basement membrane proteins) had a substantial modifying effect on disease severity and age of onset in *Lamc2^{jeb/jeb}* homozygous mice [16]. A putative modifier variant in the orthologous COL17 domain has also been suggested in a case study of JEB in humans [29]. That study also inferred the existence of additional modifiers whose identification was confounded by mixed F2 crosses and less robust phenotyping methods. The current study, utilizing multiple F2 crosses of strains showing highly discordant JEB-related phenotypes, more incisive methods, allelic matching for dominant QTL and rigorous confirmation by congenic methods was very effective in revealing such multiple QTL.

This study increases the list of known or candidate EB modifier loci in mice from 1 to 7. As summarized in Table 1, not all QTL alleles have the same direction of effect in a given strain. For example, FVB is a generally 'resistant' strain background but is concluded to contain a 'susceptible' allele on chr1 whereas B6 is generally 'susceptible' but carries a 'resistant' allele on chr11. This is not surprising as each strain phenotype represents the sum of factors which can tug in opposite directions. This agrees with previous work in which DBA1 mice carries the same 'resistant' allele as FVB at *Col17a1* but has an overall 'susceptible' phenotype [16]. It was concluded in that study that DBA1 'susceptible' loci effects must dominate phenotypically over the Col17a1 'resistant' allele, even though they could not be identified by the QTL cross.

While this study does not go as far as identifying the exact gene and mutation responsible for each QTL, it suggests strong candidates for 4 of the 6 newly identified intervals, with *Dst*

being the most strongly supported. Three, *Dst*, *Itgb4* and *Dsp*, are 'EB related genes' [3, 30]. *Dst-e* and *Itgb4* protein products dystonin-e/BP230 and integrin α6β4 are members of the hemi-desmosomal dermal-epidermal attachment complex along with laminin 332 and collagen XVII/BP180 [31, 32]. Modifier effects could be due to strengthening or weakening of structural attachment points caused by key amino acid changes [3, 33]. While mutations in desmoplakin are also known to cause EB [34, 35], highlighting its importance to dermal-epidermal connectivity, its physical location in desmosomes—relatively far away from hemidesmosomes in cellular terms—makes it harder to invoke a purely structural modifier effect. Some mutations in desmoplakin may cause EB via structural defects while other changes are structurally innocuous but have a signaling or regulatory effect [36] upon laminin 332 or other hemi-desmosomal elements. These proposed signaling changes are not physiologically noticeable in the context of normal alleles of basement membrane proteins but may have consequence in the context of a 'sensitive' EB mutation such as *Lamc2jeb/jeb*. While evoking *Dsp* as a 'strong' modifier candidate solely on the basis of it being an 'EB related gene' would be a stretch, the weak evidence for a QTL peak on chr13 in the B6 x 129 cross coupled with the strong MRL/FVB peak in the same location and the coincidence that *Dsp* not only maps to the 63Mb 95% confidence interval but also maps to the 129 = FVB/B6 = MRL<6Mb reduced candidate intervals predicted by these results greatly increases the likelihood it is the responsible gene. The fourth proposed modifier, *Ppargc1a* in the chr5 QTL, is a strong candidate because of the limited candidates remaining in its <2Mb reduced congenic interval. To date *Ppargc1a* has no known connection to EB or dermal/epidermal connectivity but its product PGC-1α has been identified as a master regulator, controlling the expression of many other genes and thereby having an impact upon multiple processes [37] including a skin specific role in melanin production [38]. It can be envisioned that PGC1A may exert a signaling/regulatory effect upon hemi-desmosomal components, perhaps akin to the proposed MMP1 effect upon collagen VII [10]. The remaining two QTL on chr6 and 10 do not contain known EB related genes. They do each contain a gene related to chr5 candidate *Ppargc1a*: *Pparg* on chr6 and *Igf1* on chr10, suggesting the *Pparg* pathway may be an important signaling modifier of dermal-epidermal connectivity processes and therefore *Lamc2jeb/jeb* based disease. Beyond those, the present large size and number of genes contained in the chr6 and 10 QTL makes attempts to surmise candidates futile. A congenic reduction approach would be a likely next step toward identifying each of these. And since the intervals do not contain genes that code for known dermal-epidermal adhesion structural components, these QTL, once identified, could reveal previously unknown structural components or, more likely, signaling/regulatory genes which impact the already known players, as the chr5 candidate apparently does.

It appears that, in terms of number of modifiers of EB, about half so far are structural (*Col17a1*, *Dst-e* and *Itgb4*) and half are 'other', with the suggestion that they are signaling or regulatory (*Dsp*, *Mmp1*, *Ppargc1a* and chr6 and 10 QTL). This is in line with a 'systems level' study of desmosomes—somewhat similar in structure and function to hemi-desmosomes- that identified 59 proteins involved of which about half were intrinsic/structural and half were accessory/regulatory [39]. In terms of modifier impact (based on alleles studied so far), *Col17a1* and *Dst-e* have much stronger effects than *Ppargc1a*, *Itgb4* and *Dsp* (based on B6129X1F$_2$ LOD plot and congenic comparisons, Figs 3B and 6A), and together dramatically alter the disease phenotype in otherwise genetically matched *Lamc2jeb/jeb* mice (Fig 6). Effect plots showing contributions of ~6-8N each for chr6, 10 and 13 (*Dsp*) QTL (Fig 4D) are somewhat less than the ~14N effect plot for chr1/*Dst-e*, and comparable to the chr5/*Ppargc1a* ~6N effect (Fig 3C), suggesting they each has somewhat less impact than *Dst-e* on phenotype. However, as shown in Fig 10, the sum effect of all FVB/MRL modifiers upon ear and tail scores exceeds that of B6/129X1 modifiers, even once *Col17a1* has been excluded. Though it appears

that tail tension results do not always fully correlate with disease observational disease scoring, they have proven to be particularly useful as a quantitative measure of dermal-epidermal adhesive strength. Because no known dermal/epidermal structural components map to the chr6 or 10 QTL, and because of the physical separation between desmosomes containing chr13 candidate desmoplakin and hemi-desmosomes and the basement membrane zone containing laminin 332, it is possible that such modifiers may involve signaling/regulatory pathways. If so, they may be more amenable to drug interventions to treat EB patients.

This work reveals QTL with a wide range of modifier effects. The considerable but incomplete delay of disease apparent when susceptible B6-*Lamc2^{jeb/jeb}* mice are provided with protective alleles for both *Col17a1^{PWD}* and the 1:22-39Mb^{129} *Dst* candidate (Fig 6). MRL and FVB display an impressive disparity in disease phenotypes (Fig 1) but, according to data shown here, do not represent the maximum possible discordance in disease states from this single JEB-nH causing *Lamc2^{jeb/jeb}* mutation. The B6.PWD congenic work suggests that a *Col17a1* AA 1275S→G change in FVB would likely be beneficial. If *Dst-e* AA 1226 and/or 1469 prove to modify, FVB shares the susceptible B6 allele at both, so converting those to the 129 allele could further increase protection. Likewise converting MRL to the B6 allele at *Col17a1* AA 1292I→T would apparently be detrimental.

Overall, these results in mice are consistent with predictions from a 589,306 genome analysis suggesting that individuals could carry primary genotypes which 'should' result in severe disease, including EB, and yet be asymptomatic, or nearly so, apparently due to genetic modifier effects [40]. These together indicate that the health fate of an individual with a particular EB causing primary mutation is not predetermined by that allele or only amenable to difficult gene or protein replacement therapy targeting the primary mutation [41]. As some individuals experience lesser symptoms due to inherited modifier alleles, it is possible that determination of beneficial signaling or regulatory effects of some of those modifiers may lead to drugs or other therapeutic treatments which can replicate those signals, leading to significant patient relief.

## Materials and methods

### Phenotyping

Males from each *Lamc2^{jeb/jeb}* strain produced and described below were tail tension tested after $CO_2$ euthanasia, usually at 10 weeks of age. Males from some strains were also aged for ear and tail scoring as previously described [16, 42]. Due to previously reported *Lamc2^{jeb/jeb}* sex phenotype differential [16, 42], all mice tested in this study were male to allow valid comparison across all genetics without sex as a confounding influence. Males were selected over females as they develop more severe disease (earlier onset) and show more dramatic and more easily interpretable disease manifestations. Data suggests testing females instead of males would not result in different QTL (not shown). Tail tension testing of groups of males at other ages was also performed and reported when informative (Fig 1). All tail tension measurements in Newtons were accompanied by censor values to note whether a partial or full tail skin 'sleeve' was removed, as is typical for *Lamc2^{jeb/jeb}* of most genotypes and ages >6 weeks old. Control mice will still record a tension value (typically ~45–75 Newtons) even though the tension device clamp slips off without removing skin [42]. This is relevant for Fig 6 results. All mouse procedures were approved by The Jackson Laboratory IACUC.

### Mice

All mice used in these studies were obtained from The Jackson Laboratory (Bar Harbor, ME). Mice homozygous for the *Lamc2^{jeb}* mutation on C57BL/6J (B6-*Lamc2^{jeb/jeb}*), 129X1/SvJ

(129X1-*Lamc2^{jeb/jeb}*, JR6859), MRL/MpJ (MRL-*Lamc2^{jeb/jeb}*, JR24120) and FVB/NJ (FVB-*Lamc2^{jeb/jeb}*, JR24337) strain backgrounds were previously described [15, 16]. *Col17a1* congenics were produced by backcrossing B6-*Lamc2^{jeb/jeb}* and MRL-*Lamc2^{jeb/jeb}* each onto FVB-*Lamc2^{jeb/jeb}* 10 generations, tracking congenic segments by PCR with SSLP markers (S1 Table). The inverse was also done, resulting in strains B6-*Lamc2^{jeb/jeb}* *Col17a1^{FVB/FVB}*, MRL-*Lamc2^{jeb/jeb}* *Col17a1^{FVB/FVB}*, FVB-*Lamc2^{jeb/jeb}* *Col17a1^{B6/B6}* and FVB-*Lamc2^{jeb/jeb}* *Col17a1^{MRL/MRL}* mice.

B6-*Lamc2^{jeb/jeb}* and 129X1-*Lamc2^{jeb/jeb}* were mated to produce $F_1$ and $F_2$ progeny as a mapping cross. 268 male B6129X1$F_2$ *Lamc2^{jeb/jeb}* homozygotes were phenotyped by tail tension test [42] at 10 weeks of age. $F_1$ males were also tension tested as controls (Fig 2). Likewise, MRL-*Lamc2^{jeb/jeb}* *Col17^{FVB/FVB}* and FVB-*Lamc2^{jeb/jeb}* were mated to produce $F_1$ and $F_2$ progeny as a second mapping cross. 176 male MRLFVB$F_2$ *Lamc2^{jeb/jeb}* *Col17a1^{FVB/FVB}* were tension tested at 15 weeks of age with $F_1$ males as controls (Fig 3). Age of tension test for each set of $F_2$ mice was chosen to optimize phenotypic differences between parental strains (Figs 1G and 2A). Due to observed large variation in size of MRLFVB$F_2$ mice, body weights were also recorded at time of tension test and used as a covariant in calculations using R/QTL.

For congenic testing, B6.129X1-*Zap70^{tm1Weis}*/J (JR4225), B6N.129S4-*Ppargc1a^{tm2.1Brsp}*/J (JR9666), B6.129P2-*Sod3^{tm1Mrkl}*/J (JR9654), B6.129S4(Cg)-*Dck^{tm1.2Radu}*/J (JR21225), B6.129S1-*Stat3^{tm1Xyfu}*/J (JR19623) and C57BL/6J-Chr 1^{A/J}/NaJ (JR4379) were obtained from The Jackson Laboratory Repository colonies as donors of chr1 (*Zap70* and A/J), chr5 (*Ppargc1a*, *Sod3* and *Dck*) and chr11 (*Stat3*) congenic segments. Extent of each congenic segment and tracking during breeding was performed by SSLP PCR using primers designed in our lab (S1 Table). Each was crossed two generations to B6-*Lamc2^{jeb/jeb}* to produce mice homozygous *Lamc2^{jeb/jeb}* and heterozygous for the desired congenic segment. These were then intercrossed to produce breeding lines homozygous *Lamc2^{jeb/jeb}* and for the desired congenic interval. When advantageous recombinants were identified, they were bred to produce separate lines homozygous for the reduced congenic interval. Extent of informative chr5 congenics tested here is shown in Fig 8C. Males homozygous *Lamc2^{jeb/jeb}* and for each lineage as indicated were tension tested at 10 weeks of age and compared to B6-*Lamc2^{jeb/jeb}* controls (Figs 5H, 8B and 9C). B6-*Lamc2^{jebjeb}* 1:22-37Mb^{129/129} was also crossed to B6(R03Q)-*Lamc2^{jeb/jeb}* *Col17a1^{PWD/PWD}* derived from previously published [16] to produce and tension test B6-*Lamc2^{jeb/jeb}* 1:22-37Mb^{129/129} *Col17a1^{PWD/PWD}* (Fig 6). Due to anticipated effects upon the immune system, *Zap70^{tm1Weis}* homozygous mice were maintained on water containing Sulfamethoxazole-Trimethoprim. A set of male B6-*Lamc2^{jeb/jeb}* was likewise raised on water containing Sulfamethoxazole-Trimethoprim and tension tested to confirm treatment did not alter results (Fig 5H).

## B6-*Lamc2^{jeb/jeb}* reduced congenic

The chr1 congenic interval bracketing *Lamc2* in B6-*Lamc2^{jeb/jeb}* previously published [16] was mapped using SSLP markers and determined to include at least 1:80-160Mb (but not as far as 64Mb or 170Mb) despite its extensive (N11) backcross. For this project, chr1 and 5 congenics crossed to this >80Mb 'long' B6-*Lamc2^{jeb/jeb}* were genotyped for 4 SSLP markers as well as *Lamc2* to ensure that, at best we could practically tell, the chr1 *Lamc2* intervals in tested congenics matched those in control mice. To reduce the congenic segment and genotyping requirements, B6-*Lamc2^{jeb/jeb}* were backcrossed to C57BL/6J two additional generations, selecting for recombinants between *Lamc2* and SSLP markers at 150Mb and 160Mb, resulting in a separate stock of B6-*Lamc2^{jeb/jeb}* with a <10Mb congenic segment available from The Jackson Laboratory as B6-*Lamc2^{jeb}*/DcrJ, JR25467 (Fig 3F). Later congenic crosses to B6-*Lamc2^{jeb/jeb}* in this series, including for chr11 and most for chr5, were made using the <10Mb

'short' congenic, and were compared to it in experiments. All other crosses, including the B6 x 129X1 F2 mapping cross and the chr1 congenics, were made using the >80Mb 'long' congenic, and were compared to it in experiments. The <10Mb congenic is publicly available as JAX Repository strain JR25467.

## Genotyping

DNA was collected from mice requiring genotyping, including all $F_2$ mapping mice, via retro-orbital blood extraction followed by washes in Buffone's Buffer and Proteinase K digestion. All genotyping was done by SSLP PCR using primers designed in Primer Express 2.0 software (Applied Biosystems, Inc.) to bracket regions of dinucleotide repeats, using annealing temperatures of 58–60°C and giving product sizes of ~100 bp. SSLP PCR used 40 cycles of 94°C for 30 seconds, 60°C for 60 seconds and 70°C for 60 seconds. Band resolution was performed using 0.7% agarose, 1.5% Synergel (both from BioExpress) and 150 ng/ml ethidium bromide gels in TAE buffer. Primers used are listed in S1 Table.

268 male B6129X1F$_2$-*Lamc2^{jeb/jeb}* mice were generated and tension tested at 10 weeks of age. A subset of 127 were genotyped for 88 SSLP markers across the genome (S1 Table). T-test comparisons in Microsoft (MS) Excel of tension sorted by genotype at each marker (B6 vs het, 129 vs het, B6 vs 129) were used to get a first estimate of significance. SSLP markers with at least suggested significance were tested for the remainder of the 268 mice prior to R/QTL analysis. All 176 tension tested MRLFVBF$_2$-*Lamc2^{jeb/jeb}* males were tested with 89 SSLP markers spread across the genome prior to R/QTL analysis (S1 Table). All congenics crossed were typed with appropriate SSLP markers to ensure inclusion or exclusion of desired intervals and to detect recombinants. All recombinants were treated as genetically different and were often bred to generate new lineages.

## Survival

All mouse information for our laboratory is tracked in JAX Colony Management System (JCMS, The Jackson Laboratory, Bar Harbor, ME) software. For Fig 2D, data for all male mice from three strains in JCMS (whether experimental, breeding or other) were extracted, age of death in weeks was calculated and values were assigned based on 'cause of death' indicating whether or not the age of death is censored. Data was imported to GraphPad PRISM for graphing and statistical calculation. N = 1462 FVB males, 307 FVB.B6 males and 327 FVB.MRL males.

## Percent polymorphism plots

A copy of the entire Center for Genome Dynamics Mouse Diversity Array 1 (CGD-MDA1) data set containing over 470,000 SNP genotypes for 142 inbred strains [43] was downloaded by chromosome (chr) as csv files from Mouse Phenome Database (MPD). The chr files were restricted in MS Excel to just the 314,349 markers which had informative alleles for all 142 strains. These was imported to a single table in Microsoft Access after minor required format editing of strain names. A query of this table (AllChr Query) was designed to pull up only chromosome, bp location, a conversion of location to Mb including only one decimal place for grouping (Mb: (Int([bploc]/100000))/10), alleles for four selected strains (i.e. strain1: C57BL/6J), defined comparisons of those strains 1vs2, 3vs4, 13vs24, 1v2NOT3v4 and 3v4NOT1v2 (i.e. 13vs24: IIf([strain1] = [strain3],IIf([strain2] = [strain4],IIf([strain1] = [strain2],0,1),0),0)) and selection of which comparison to forward for further analysis (i.e. Comp: [3v4NOT1v2]). A second query (AllChr Query2) looks at the first and reports only Chr, Mb, count of Mb, sum of 1vs2, sum of 3vs4 and sum of Comp, thus reporting only one line result per 0.1Mb interval

across the genome. Third level queries filtered to include single chromosomes (i.e. Chr = 1) and sometimes only regions of chromosomes (i.e. Mb >22 and <39) then divide SumOf1vs2, SumOf3vs4 and SumOfComp by 'Mb count' to produce polymorphic percentages, which were then graphed in MS Access reports as histograms by 0.1Mb interval in Figs 5A–5F, 7, 8A and 9A. 129X1/SvJ and 129S1/SvImJ are related strains but have significant regions of disparate genetic heritage [44]. 129S1 has been fully sequenced and is available for detailed comparison to B6 reference in the Sanger Mouse Genomes Project (sanger.ac.uk/sanger/Mouse_Snp-Viewer/; site is no longer available, but data can be accessed at GenomeMUSter Search, https://mpd.jax.org/genotypes) while 129X1 used in this study has not. 129X1 vs 129S1 percent polymorphic plot comparisons were performed here to determine if they shared strain heritage in regions of interest so 129S1 could fairly be used as a substitute for 129X1 in comparing to B6 [20].

## Other data mining

Aside from above, the MPD CGD-MDA1 SNP dataset was surveyed for C57BL/6J (B6), 129X1/SvJ, FVB/NJ and MRL/MpJ alleles across specific candidate intervals 5:50.5–52.366Mb, 11:115.8–116.2Mb and 13:38–38.3Mb and candidate genes *Dst*, *Ppargc1a*, *Itgb4*, *Dsp*, *Pparg* and *Igf1* for total number of SNPs and number of polymorphic SNPs between particular strain combinations to plot localized percent polymorphic (Figs 5G, 8E and 9B). Only SNPs with allelic information available for all 4 strains were used. Since congenic donors involved several 129 strains (129X1, 129P4, 129S1 and 129S4), the MPD Broad2 dataset was surveyed for all 129 substrains for the chr5 and 11 congenic intervals to ensure shared strain heritage prior to obtaining each strain.

Sanger Mouse Genomes Project was used to compare B6 reference sequence to that of 129X1 surrogate strain 129S1 (based on CGD-MDA1) to determine likely B6 vs 129X1 polymorphisms for candidate regions and genes. Comparisons used Sanger release REL-1505 except *Dst*, which used REL-1410. Strains selected are 129S1 and occasionally A/J and/or FVB. 'SNP/Indel type' selections for searches are Sanger default (all except intergenic variant, intron variant and unspecified) unless specified in Results. The Sanger online Query has been retired since this work was performed. It refers future queries to https://www.informatics.jax.org/snp. Strain SNP comparisons were also performed using GenomeMUSter https://mpd.jax.org/genotypes.

Ensembl was queried for gene lists in various intervals of interest and for *Dst* transcript comparison. All mouse Ensembl references are to NCBI Build 38. BioGPS (biogps.org) was also surveyed for tissue expression patterns of genes of interest, with emphasis on high expression in mouse epidermis and cornea and human bronchial epithelial cells typical of 'EB related genes'.

## Statistics

Tension comparisons are made using ANOVA in GraphPad PRISM. Ear and tail score statistics are based on age at which scores first reached a threshold of '4' ('moderately affected', censored values for mice which did not reach '4' before euthanized), survival compared using both Log-rank (Mantel-Cox) and Gehan-Breslow-Wilcoxon tests in Prism. Reported values are the less significant of the two. Statistics for the survival curve in Fig 2 are Log-Rank in PRISM. QTL mapping was performed using R/QTL [18, 19] 1000 permutations were performed to determine the genome-wide thresholds for QTL detection [45]. Four thresholds 1%, 5%, 10% and 63% were calculated from the permutation results. QTL with LOD score above the 10% threshold are considered as significant QTL, while those above 63% but below 10%

threshold are considered to be suggestive QTL [46]. Following genome-wide scans, we also performed pair-scans for all pairs of QTL locations on each chromosome. As pair-scans permutations are extremely resource intensive, genome-wide two-dimensional scan significance thresholds were based on the 5% thresholds suggested by Broman and Sen [18, 19]. Specifically, for a mouse intercross the following thresholds were used $(T_f, T_{fv1}, T_i, T_a, T_{av1}) = (9.1, 7.1, 6.3, 6.3, 3.3)$. Following genome-wide single scans and pair-scans, all QTL along with possible QTL x QTL interaction identified from a single QTL scan and pair-scan were fit into multiple regression models in the presence of significant covariates, if any. By doing so, variations of the phenotype in the models were estimated. P values for terms in the multiple regression model were calculated. Terms were dropped sequentially until all of the terms in the model were significant at 5% level for both the main QTL effects and their interaction effects. R/QTL values were given in cM. Conversion to Mb positions reported here was done using Mouse Map Converter (http://cgd.jax.org/mousemapconverter/) Sex-Averaged cM—Cox to GRCm38 Mb.

## Supporting information

**S1 Table. Polymorphic SSLP markers used in this study.**
(XLSX)

**S1 Data.**
(XLSX)

## Author Contributions

**Conceptualization:** Thomas J. Sproule, Derry C. Roopenian, John P. Sundberg.

**Data curation:** Thomas J. Sproule.

**Formal analysis:** Thomas J. Sproule, Vivek M. Philip, Nabig A. Chaudhry.

**Funding acquisition:** Derry C. Roopenian, John P. Sundberg.

**Investigation:** Thomas J. Sproule.

**Methodology:** Thomas J. Sproule.

**Project administration:** Derry C. Roopenian, John P. Sundberg.

**Writing – original draft:** Thomas J. Sproule, Derry C. Roopenian, John P. Sundberg.

**Writing – review & editing:** Thomas J. Sproule, Vivek M. Philip, Nabig A. Chaudhry, Derry C. Roopenian, John P. Sundberg.

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
