## [Decision Letter · Decision Letter 0]

9 May 2023

PONE-D-23-09055Seven naturally variant loci serve as genetic modifiers of Lamc2jeb induced non-Herlitz Junctional Epidermolysis Bullosa in micePLOS ONE

Dear Dr. Sproule,

Thank you for submitting your manuscript to PLOS ONE. After careful consideration, we feel that it has merit but does not fully meet PLOS ONE’s publication criteria as it currently stands. Therefore, we invite you to submit a revised version of the manuscript that addresses the points raised during the review process.

The manuscript is in very good shape and the reviewers only had minor concerns with writing and asked for a few clarifications.

Please respond to those carefully, and review the writing and formatting (particularly of supplements and figure legends) to tidy up any potential typos or errors.

We look forward to receiving your revised manuscript.

Kind regards,

Arnar Palsson, Ph.D.

Academic Editor

PLOS ONE

Journal Requirements:

2. We noted in your submission details that a portion of your manuscript may have been presented or published elsewhere. [Data from inbred strains (B6, 129, MRL, FVB) homozygous Lamc2jeb/jeb used as controls in this manuscript for congenics and F2 crosses (also Lamc2jeb/jeb) was mostly previously published as new findings in PMID 24550734. Inclusion of that information is necessary to make comparisons.] Please clarify whether this [conference proceeding or publication] was peer-reviewed and formally published. If this work was previously peer-reviewed and published, in the cover letter please provide the reason that this work does not constitute dual publication and should be included in the current manuscript.

“This work was supported by grants from DEBRA Austria, DEBRA UK, and The Jackson Laboratory.”

“DCR received awards 'Roopenian 1' from Debra UK (https://www.debra.org.uk/) and 'Roopenian 2' from Debra Austria (https://www.debra-austria.org/). The Jackson Laboratory (jax.org) provided supplemental support for these studies. The funders had no role in study design, data collection and analysis, decision to publish, or preparation of the manuscript.”

Additional Editor Comments (if provided):

The manuscript is in very good shape and the reviewers only had minor concerns with writing and asked for a few clarifications.

Please respond to those carefully, and review the writing and formatting (particularly of supplements and figure legends) to tidy up any potential typos or errors.

Reviewers' comments:

Reviewer's Responses to Questions

**Comments to the Author**

1. Is the manuscript technically sound, and do the data support the conclusions?

Reviewer #1: Yes

Reviewer #2: Yes

2. Has the statistical analysis been performed appropriately and rigorously? 

Reviewer #1: Yes

Reviewer #2: Yes

3. Have the authors made all data underlying the findings in their manuscript fully available?

Reviewer #1: Yes

Reviewer #2: Yes

4. Is the manuscript presented in an intelligible fashion and written in standard English?

Reviewer #1: Yes

Reviewer #2: Yes

5. Review Comments to the Author

Reviewer #1: In this manuscript, the authors present a highly rigorous analysis of modifier QTL in a mouse model of Epidermolysis Bullosa (EB). Expanding on their 2014 PLoS Genetics publication that identified Col17a1 as the first definitive genetic modifier of EB, the authors present six additional QTL that, while apparently inconsequential in wild-type mice, significantly affect EB phenotypic severity in the context of homozygous, hypomorphic Lamc2 mutations. While causal variants are not definitively identified in the current study, likely candidates are presented based on several convincing lines of evidence. The manuscript is well written, and conclusions are justified as stated, though some sections may be challenging for non-experts to understand due to the complexity of genetic analyses. Overall, the study warrants publication in PLoS One. A few minor modifications/clarifications, as detailed below, may improve overall clarity of the manuscript.

General comments:

Some aspects of this study are quite complex, involving multiple distinct strains with important though not always straightforward genetic differences. It may be helpful to include a summary Table or Figure that details predicted (or known) genotypes for each QTL along with predicted effector gene(s).

In the final two sentences of the Discussion section, the authors provide solid rationale for how the data obtained in this study could be applied translationally to improve patient outcomes. It would be beneficial to include this rationale in the Introduction and/or Abstract as well to highlight its importance.

Specific comments:

Page 4, Line 75: The abbreviation "LTA" is not previously defined and is not obvious.

Page 9, Lines 166-167: It would be helpful to clarify that the short congenic strain was generated as part of the current study.

Page 11, Lines 192-194: Inclusion of body weight as a covariant may be appropriate, but it is not immediately clear how this variable would influence tail tension test results or why it was considered as a potential covariant, aside from the substantial variation observed. If inclusion as a covariant affects QTL significance for this cohort, is it appropriate to apply it to others as well (even though effects may be less)? Perhaps a bit more explanation would clarify matters for readers.

Page 14, Line 230: The authors refer to a specific dystonin isoform (Dst-e) as a modifier candidate. More background should be provided on dystonin isoforms and why this one was specifically mentioned (presumably due to expression patterns?).

Figure 3A: It is interesting that several, perhaps the majority, of F2 mice exhibit a tail tension phenotype indicative of lower disease severity (higher tension) than either of the parental strains. What is the interpretation of this result?

Figure 3B: The extremely high LOD score for the chr1 QTL makes it somewhat difficult to appreciate the significance of other QTL. It may be beneficial to include a sub-panel with a y-axis similar to that of Fig. 4B-C or use a staggered y-axis.

Reviewer #2: Epidermolysis Bullosa (EB) is a group of skin disorders caused by genetic mutations, several of which have been described in literature. Published genetic mutations were able to account for all subtypes of EB, but there still remains clinical variations within each subtypes that were not explained by the primary mutations. These phenotypic variations within the same primary mutations may be caused by enviromental factors or other genetic mutations. This manuscript dissects genetic modifiers that may contribute to the severity and clinical variations found in EB by using multiple F2 crosses of strains that are homozygotes for Lamc2jeb/jeb, which produces insufficient healthy laminin 332.

The present study reports six novel QTLs that were proposed to play a role in the clinical outcome of EB. For each QTL, the authors investigate the genomic area identified by the 95% confidence interval for missense SNPs/Indels. The authors used genetic variants in related additional crosses to reduce genomic regions of interest. Overall, the study was performed using appropriate experimental and statistical methods, and conclusions were supported based on the presented data.

Minor issues that authors should consider to address are:

1. Please provide the percentage of phenotypic variations (Tensions) explained by each QTL reported, which will help readers put into prespective the impact of each QTL on the outcome of EB.

2. The third effect plots in Fig 3C showed that mouse carrying allele BB has the highest tension, unlike the other three effect plots in the same figure. Could you please provide explanations.

3. This study identifies potential DNA variants (SNPs/Indels) that may regulate EB outcomes, which will be useful for other researchers. Please provide a list of SNPs/Indels of interest as in Fig. 5J that were identified for the rest of QTLs.

6. PLOS authors have the option to publish the peer review history of their article (what does this mean?). If published, this will include your full peer review and any attached files.

Reviewer #1: No

Reviewer #2: No

---

## [Author Response · Author response to Decision Letter 0]

13 Jun 2023

Response to Reviewers

Thank you for your thoughtful comments concerning our submitted manuscript Seven naturally variant loci serve as genetic modifiers of Lamc2jeb induced non-Herlitz Junctional Epidermolysis Bullosa in mice. 

Reviewer 1.

General comments: 

“It may be helpful to include a summary Table or Figure that details predicted (or known) genotypes for each QTL along with predicted effector gene(s).”

Response: Table 1 has been added to summarize the modifier effects of all QTL and the total effects for each of the four strains used in this study: 129X1, B6, FVB and MRL.

“In the final two sentences of the Discussion section, the authors provide solid rationale for how the data obtained in this study could be applied translationally to improve patient outcomes. It would be beneficial to include this rationale in the Introduction and/or Abstract as well to highlight its importance.”

Response: The overall intent of this manuscript was to communicate the importance of multiple genetic modifiers in determining the severity of EB. Those with the strongest effect were related to structural proteins, while others showing a weaker but significant effects were consistent with regulatory/signaling proteins. We think it would be premature to overstress the therapeutic potential of targeting such proteins at this point. 

Specific comments:

“Page 4, Line 75: The abbreviation "LTA" is not previously defined and is not obvious.”

Response: We have amended the text (LTA=Lymphotoxin alpha). 

“Page 9, Lines 166-167: It would be helpful to clarify that the short congenic strain was generated as part of the current study.”

Response: In page 9 we have clarified that the B6-Lamc2jeb/jeb short congenic was generated as part of this study.

“Page 11, Lines 192-194: Inclusion of body weight as a covariant may be appropriate, but it is not immediately clear how this variable would influence tail tension test results or why it was considered as a potential covariant, aside from the substantial variation observed.”

Response: JEB, like other forms of EB, manifests not only in the skin but in other epithelialized surfaces including the airways and the digestive tract. The Lamc2 mutation results in systemic effects including failure to thrive and shortened lifespans. While we think it unlikely that body weight would directly affect tail tension values (and vice versa), it is reasonable to expect that body size/weight may covary with tail tension measurements. We have thus revised page 11 accordingly.

“Page 14, Line 230: The authors refer to a specific dystonin isoform (Dst-e) as a modifier candidate. More background should be provided on dystonin isoforms and why this one was specifically mentioned (presumably due to expression patterns?).”

Response: We have amended the text to clarify this concern.

“Figure 3A: It is interesting that several, perhaps the majority, of F2 mice exhibit a tail tension phenotype indicative of lower disease severity (higher tension) than either of the parental strains. What is the interpretation of this result?”

Response: The large spread in the 129XB6 F2 cohort is not unexpected given the number of F2 mice analyzed. The fact that the F2 mice were skewed towards higher tension values may be explained by the underlying genetics of that cross. The chr13 QTL appears to be weak for B6 vs 129X1, but chrs 11 and 13 are two QTL in which het phenotype matches the 129X1 phenotype (one allele dominant rather than typical semi-dominant). If chr13 was an important player and ¾ of F2s include its dominant phenotype because either BX or XX, that could explain an increase in resistant mice. 

“Figure 3B: The extremely high LOD score for the chr1 QTL makes it somewhat difficult to appreciate the significance of other QTL. It may be beneficial to include a sub-panel with a y-axis similar to that of Fig. 4B-C or use a staggered y-axis.”

Response: A new panel 3C has been added to accomplish this.

Reviewer 2. 

“Please provide the percentage of phenotypic variations (Tensions) explained by each QTL reported, which will help readers put into perspective the impact of each QTL on the outcome of EB.”

Response: Percentage variation has been added to newly created Table 1. 

“The third effect plots in Fig 3C showed that mouse carrying allele BB has the highest tension, unlike the other three effect plots in the same figure. Could you please provide explanations.”

Response: New Table 1 summarizes this nicely and comments/explanations have been added to the Discussion.

“This study identifies potential DNA variants (SNPs/Indels) that may regulate EB outcomes, which will be useful for other researchers. Please provide a list of SNPs/Indels of interest as in Fig. 5J that were identified for the rest of QTLs.” 

Response: A list for chr5/Ppargc1a candidate SNPs has been added as Fig 8F. Lists of chr11/Itg4b and chr13/Dsp candidates have been added as new Table 2. Additional comments have been added to the manuscript to clarify that the chr6 and chr10 candidate genes Pparg and Igf1 (which are only listed because of their relationship to Ppargc1a – the chr6 and chr10 regions are large and contain many candidates though none clearly jump out as better than others) do not carry any SNPs with predicted allelic pattern FVB / MRL and B6 = 129X1 and good candidate functional consequences, such as missense or splice region variants. Therefore no chr6 or chr10 SNP candidates are offered in Table 2.

---

## [Decision Letter · Decision Letter 1]

22 Jun 2023

Seven naturally variant loci serve as genetic modifiers of Lamc2jeb induced non-Herlitz Junctional Epidermolysis Bullosa in mice

PONE-D-23-09055R1

Dear Dr. Sproule,

We’re pleased to inform you that your manuscript has been judged scientifically suitable for publication and will be formally accepted for publication once it meets all outstanding technical requirements.

Kind regards,

Arnar Palsson, Ph.D.

Academic Editor

PLOS ONE

Additional Editor Comments (optional):

Reviewers' comments:

Reviewer's Responses to Questions

**Comments to the Author**

1. If the authors have adequately addressed your comments raised in a previous round of review and you feel that this manuscript is now acceptable for publication, you may indicate that here to bypass the “Comments to the Author” section, enter your conflict of interest statement in the “Confidential to Editor” section, and submit your "Accept" recommendation.

Reviewer #1: All comments have been addressed

Reviewer #2: All comments have been addressed

2. Is the manuscript technically sound, and do the data support the conclusions?

Reviewer #1: (No Response)

Reviewer #2: Yes

3. Has the statistical analysis been performed appropriately and rigorously? 

Reviewer #1: (No Response)

Reviewer #2: Yes

4. Have the authors made all data underlying the findings in their manuscript fully available?

Reviewer #1: (No Response)

Reviewer #2: Yes

5. Is the manuscript presented in an intelligible fashion and written in standard English?

Reviewer #1: (No Response)

Reviewer #2: Yes

6. Review Comments to the Author

Reviewer #1: (No Response)

Reviewer #2: (No Response)

7. PLOS authors have the option to publish the peer review history of their article (what does this mean?). If published, this will include your full peer review and any attached files.

Reviewer #1: No

Reviewer #2: No

---

## [Editor Report · Acceptance letter]

4 Jul 2023

PONE-D-23-09055R1 

Seven naturally variant loci serve as genetic modifiers of *Lamc2^jeb^* induced non-Herlitz Junctional Epidermolysis Bullosa in mice 

Dear Dr. Sproule:

I'm pleased to inform you that your manuscript has been deemed suitable for publication in PLOS ONE. Congratulations! Your manuscript is now with our production department. 

Kind regards, 

on behalf of

Dr. Arnar Palsson 

Academic Editor

PLOS ONE